# GPR101 drives growth hormone hypersecretion and gigantism in mice via constitutive activation of $G_s$ and $G_{q/11}$

Dayana Abboud [1], Adrian F. Daly [2], Nadine Dupuis[1], Mohamed Ali Bahri[3], Asuka Inoue[4], Andy Chevigné [5], Fabien Ectors[6], Alain Plenevaux[3], Bernard Pirotte [7], Albert Beckers [2✉] & Julien Hanson [1,7✉]

Growth hormone (GH) is a key modulator of growth and GH over-secretion can lead to gigantism. One form is X-linked acrogigantism (X-LAG), in which infants develop GH-secreting pituitary tumors over-expressing the orphan G-protein coupled receptor, GPR101. The role of GPR101 in GH secretion remains obscure. We studied GPR101 signaling pathways and their effects in HEK293 and rat pituitary GH3 cell lines, human tumors and in transgenic mice with elevated somatotrope Gpr101 expression driven by the rat *Ghrhr* promoter (*Ghrhr*^*Gpr101*). Here, we report that Gpr101 causes elevated GH/prolactin secretion in transgenic *Ghrhr*^*Gpr101* mice but without hyperplasia/tumorigenesis. We show that GPR101 constitutively activates not only $G_s$, but also $G_{q/11}$ and $G_{12/13}$, which leads to GH secretion but not proliferation. These signatures of GPR101 signaling, notably PKC activation, are also present in human pituitary tumors with high GPR101 expression. These results underline a role for GPR101 in the regulation of somatotrope axis function.

[1] Laboratory of Molecular Pharmacology, GIGA-Molecular Biology of Diseases, University of Liège, Liège, Belgium. [2] Department of Endocrinology, Centre Hospitalier Universitaire de Liège, University of Liège, Liège, Belgium. [3] GIGA-CRC in vivo Imaging, University of Liège, Liège, Belgium. [4] Graduate School of Pharmaceutical Sciences, Tohoku University, Sendai, Miyagi, Japan. [5] Immuno-Pharmacology and Interactomics, Department of Infection and Immunity, Luxembourg Institute of Health, Esch-sur-Alzette, Luxembourg. [6] GIGA—Transgenics Platform, Liège University, Liège, Belgium. [7] Laboratory of Medicinal Chemistry, Centre for Interdisciplinary Research on Medicines (CIRM), University of Liège, Liège, Belgium. ✉email: albert.beckers@uliege.be; j.hanson@uliege.be

Normal physical growth is a fundamental biological process that integrates a multitude of signals from hormonal effectors, modulators, and target tissues[1]. While immensely complex, growth is also tightly regulated and highly choreographed to balance energy acquisition and expenditure with the evolving metabolic needs of the growing body[1]. Growth hormone (GH) is the archetypical modulator of growth across many species and it has pleiotropic effects on organ metabolism, body composition and growth, either directly or through factors like insulin-like growth factor 1 (IGF-1)[2].

As a potent hormonal, growth-enhancing pathway, the GH-IGF-1 axis exists within a canonical network of modulators acting directly and indirectly on the pituitary cells secreting GH, the somatotropes. The main regulators from the hypothalamus are the stimulatory GH releasing hormone (GHRH) and the inhibitor somatostatin (SST), which act via their receptors at the level of the somatotrope to regulate GH synthesis and secretion. Peripheral levels of IGF-1 also have an important feedback on GH secretion from the pituitary, while other hormonal factors like ghrelin (secreted by the stomach) act to modulate GH secretion in relation to feeding[3].

Pathological dysregulation of GH axis function is rare, due to its central role in maintaining normal body size, composition and metabolism. The most severe form of GH-related overactivity is pituitary gigantism, where somatotrope tumorigenesis leads to chronic GH hypersecretion during childhood/adolescence[4]. Pituitary gigantism can lead to catastrophic overgrowth of long bones and severe disease effects due to overgrowth of multiple organ systems[4]. Nearly half of pituitary gigantism cases are due to established genetic causes, the most severe form being X-linked acrogigantism (X-LAG), which is implicated in many of the tallest humans with pituitary gigantism[5–8]. X-LAG is characterized by infant-onset somatotrope tumors and hyperplasia that produce high levels of GH and prolactin (PRL), due to genomic rearrangements on chromosome Xq26.3 leading to duplications involving the GPR101 gene. GPR101 is highly over-expressed in X-LAG tumors as compared with normal pituitary[7,9].

GPR101 is a G-protein-coupled receptor (GPCR) that is constitutively coupled to $G_s$ and has no known ligand and is therefore an orphan GPCR[10–12]. It is expressed at high levels in regions of the hypothalamus, the nucleus accumbens and in the fetal pituitary during somatotrope development and maturation[12,13]. To better understand the place of GPR101 in somatotrope development and regulation, we develop herein a transgenic mouse model ($Ghrhr^{Gpr101}$) that expresses the murine ortholog of the receptor (Gpr101) under the control of the Ghrhr promoter. This construction drives the expression of the transgene in the terminally differentiated somatotropes and somatomammotropes of the POU domain, class 1, transcription factor 1 (POU1F1), also named Pituitary-specific positive transcription factor 1 (Pit-1), lineage[14,15]. This pituitary-specific Gpr101 overexpression in mice leads to a gigantism phenotype characterized by skeletal overgrowth accompanied by elevated GH, IGF-1, and PRL secretion. Chronic GH/IGF-1 hypersecretion in transgenic mice associates with classical metabolic effects of elevated glucose levels, decreased fat mass and increased lean mass[16]. Crucially, this GH hypersecretion occurs in the absence of pituitary hyperplasia or tumorigenesis, indicating that the role of Gpr101 in the pituitary enhances secretion rather than enhancing proliferation. In addition, we find that GPR101-induced GH secretion is dependent on $G_s$ and $G_{q/11}$ pathways, notably through the activation of Protein kinase A (PKA) and Protein kinase C (PKC). We validate these findings in transgenic mice and observe that the pituitary adenomas of X-LAG patients, that are characterized by high expression levels of GPR101[7], have an increase of PKC activity compared to other GH-secreting tumors. GPR101 is a constitutively active GPCR coupled to multiple G proteins that acts via $G_s$- and $G_{q/11}$-dependent pathways to promote hormonal activity of the somatotrope axis.

## Results

**Gpr101 in the pituitary promotes GH secretion in vivo.** To investigate the impact of GPR101 signaling on somatotrope function, we generated a mouse model ($Ghrhr^{Gpr101}$) expressing Gpr101 under the control of the rat Ghrhr promoter, which drives expression in terminally differentiated somatotropes and somatomammototropes of the POU1F1/Pit-1 lineage[14,15,17,18]. The rat Ghrhr promoter was fused with FLAG-Gpr101 coding sequence and the linearized construct (Supplementary Fig. 1a) was injected into fertilized mouse oocytes. We obtained several founders that incorporated the transgene (Supplementary Figs. 1a, b, 3) and showed expression of FLAG-tagged Gpr101 at the membrane of pituitary somatotropes and somatomammotropes, as assessed by FLAG-staining (Supplementary Fig. 1c) and colocalization with Ghrhr, Pit-1, GH (Fig. 1a–c), and PRL (Supplementary Fig. 1k). The mRNA transcripts for the transgene were also detected in embryos, juvenile, and adult mouse pituitaries (Supplementary Fig. 1e, f). We did not find transgene expression in other brain structures, especially the hypothalamus (Supplementary Fig. 1g, h). The expression of the protein could be detected at embryonic day 16 (E16.5) (Supplementary Fig. 1i). FLAG-Gpr101 did not co-stain with the progenitor marker Sox2, suggesting it was present only in terminally differentiated cells (Supplementary Fig. 1j)[19]. Our transgene was not found to be expressed in corticotropes, gonadotropes, or thyrotropes (Supplementary Fig. 1l–n).

Our initial step was to characterize the activity of the somatotrope axis in the $Ghrhr^{Gpr101}$ mice versus controls. We monitored the plasma levels of GH and IGF-1 at different time points and found that even at the earliest time-point (6 weeks), the transgenic (Tg) mice had elevated GH and IGF-1 levels (Fig. 1d, e). As expected, the GH levels decreased with age but they remained elevated in the Tg lines, in both males and females (Fig. 1d). IGF-1 remained consistently increased up to the age of 52 weeks (Fig. 1e). We also observed hyperprolactinemia, in both males and females (Fig. 1f), likely due to the presence of the transgene in somatomammotropes (Supplementary Fig. 1k). We followed the growth of male and female mice from 3 to 69 weeks (Fig. 1g, h). The elevated circulating levels of GH and IGF-1 translated into a significantly increased body length (nose-to-anus, the tail length being unaffected, Supplementary Fig. 2a) in the $Ghrhr^{Gpr101}$ mice after 24 weeks of age and was more pronounced after 1 year (Fig. 1g–j). Despite the increased growth of the $Ghrhr^{Gpr101}$ mice, no significant differences occurred versus wild-type (WT) in terms of body weight (Supplementary Fig. 2c). However, there were extensive skeletal changes involving both the axial skeleton and long bones (Fig. 1k). Femoral and tibial length was increased in the $Ghrhr^{Gpr101}$ mice as compared with controls (Fig. 1l, m and Supplementary Fig. 2e). Other bones displayed no statistically significant differences between WT and Tg animals (Supplementary Fig. 2b). Chronic GH/IGF-1 hypersecretion has a series of well-established effects on metabolism and body composition and these were present in the $Ghrhr^{Gpr101}$ mice. As compared with WT, the $Ghrhr^{Gpr101}$ mice of both sexes had significantly lower fat mass (as illustrated with epididymal white fat, Fig. 2a), while many organ weights were not significantly altered (Supplementary Fig. 2d). These effects of lowered fat mass were clearly visualized and quantified on whole body CT images, as illustrated in Fig. 2b, c. In parallel, lean mass determined by CT-scan was elevated due to chronic GH/IGF-1 secretion in the $Ghrhr^{Gpr101}$ versus WT (Fig. 2d).

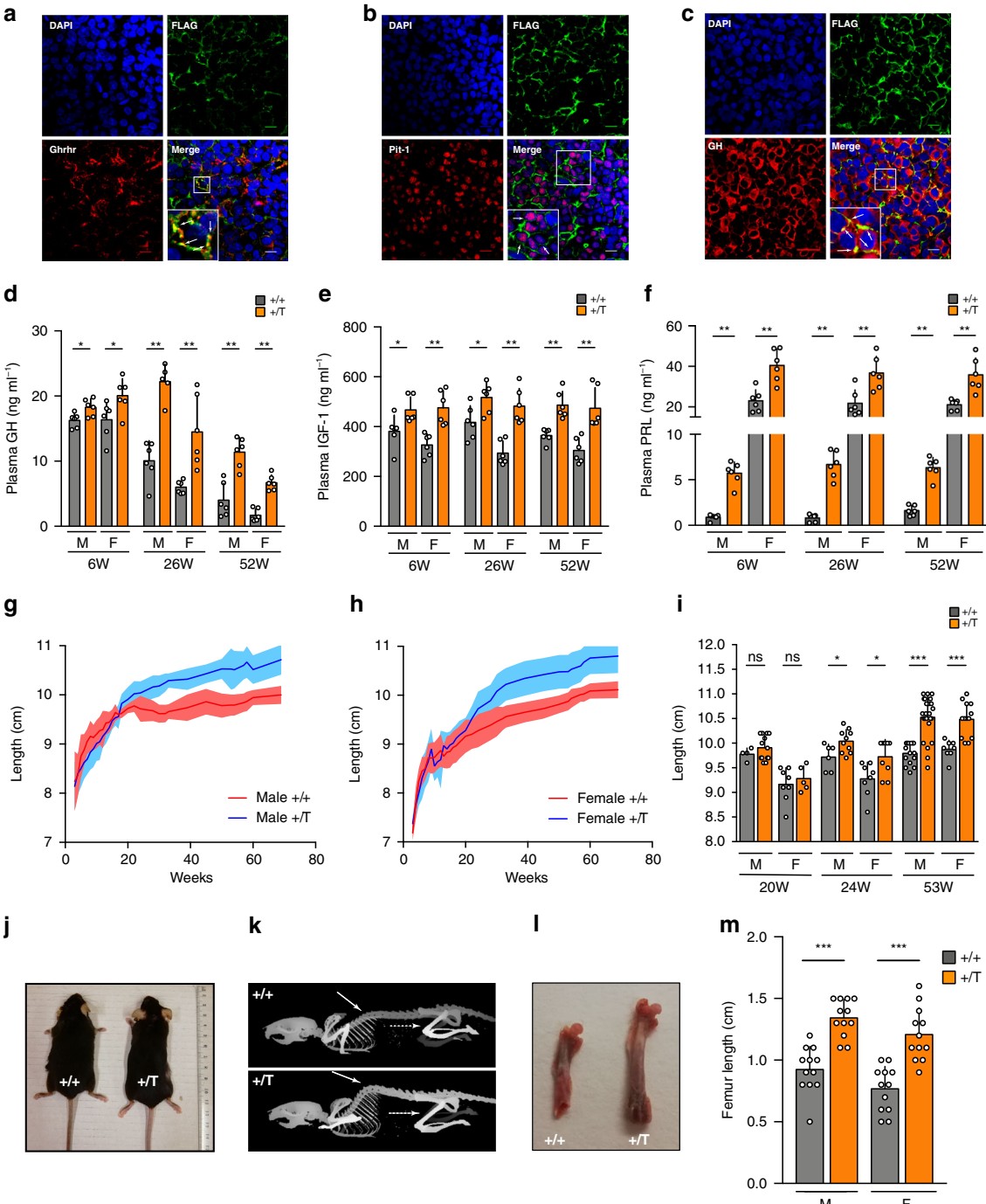

**Fig. 1 Gpr101 promotes GH/IGF-1 and PRL hypersecretion and overgrowth in vivo. a–c** Immunofluorescent staining of anterior pituitary from 29-week-old *Ghrhr*^*Gpr101*^ Tg mice. Blue: DAPI. Green: FLAG antibody. Red: **a** Ghrhr antibody, **b** Pit-1 antibody and **c**. GH antibody. (×60 magnification, scale bar: 10 μm). These experiments were repeated at least 3 times. **d**, **e** Determination in WT (+/+) and *Ghrhr*^*Gpr101*^ (+/T) (*n* = 6 mice per group) of plasma levels of **d** GH. Males 6 W: *p* = 0.0411, 26 W and 52 W: *p* = 0.0022. Females 6, 26, and 52 W: *p* = 0.0022. **e** IGF-1. Males 6 W and 26 W: *p* = 0.0260, 52 W: *p* = 0.0022. Females 6, 26, and 52 W: *p* = 0.0022. **f** PRL. Males 6, 26, and 52 W: *p* = 0.0022. Females 6 W: *p* = 0.0043, 26 W: *p* = 0.0087 52 W: *p* = 0.0022. **g**, **h** Growth curves (length, nose-to-anus) of WT (+/+, males *n* = 4–18 mice, females *n* = 6–16 mice) and *Ghrhr*^*Gpr101*^ (+/T, males *n* = 5–24 mice, females *n* = 5–13 mice) between week 3 and week 69. **i** Quantification and statistical analysis of the lengths of mice at different time points. Males 20 W: *n* = 4 (+/+) and 13 (+/T) mice, *p* = 0.4790; 24 W: *n* = 6 (+/+) and 10 (+/T) mice, *p* = 0.0493; 53 W: *n* = 15 (+/+) and 24 (+/T) mice, *p* = 0.0001. Females 20 W: *n* = 8 (+/+) and 5 (+/T) mice, *p* = 0.6169; 24 W: *n* = 8 (+/+) and 9 (+/T) mice, *p* = 0.0349; 53 W: *n* = 8 (+/+) and 13 (+/T) mice, *p* = 0.0002. **j** Macroscopic findings regarding body length of WT (+/+) and *Ghrhr*^*Gpr101*^ (+/T) aged 53 weeks. **k** CT images of WT (+/+) and *Ghrhr*^*Gpr101*^ (+/T) mice (age 27 weeks). Plain arrow indicates skeletal kyphosis and dashed arrow indicates the femur (*n* = 4–5 mice per group). **l**. Extracted femurs of 27-weeks-old WT (+/+) and *Ghrhr*^*Gpr101*^ (+/T) mice. **m** Quantification of femur length (*n* = 12 femurs from 6 mice per group, *p* = 0.0001 for Males and Females). All Data are Mean ± S.D. For statistical analysis of all data, a two-sided Mann–Whitney test was used. ns not significantly different; *\*p* < 0.05; *\*\*p* < 0.01 *\*\*\*p* < 0.001.

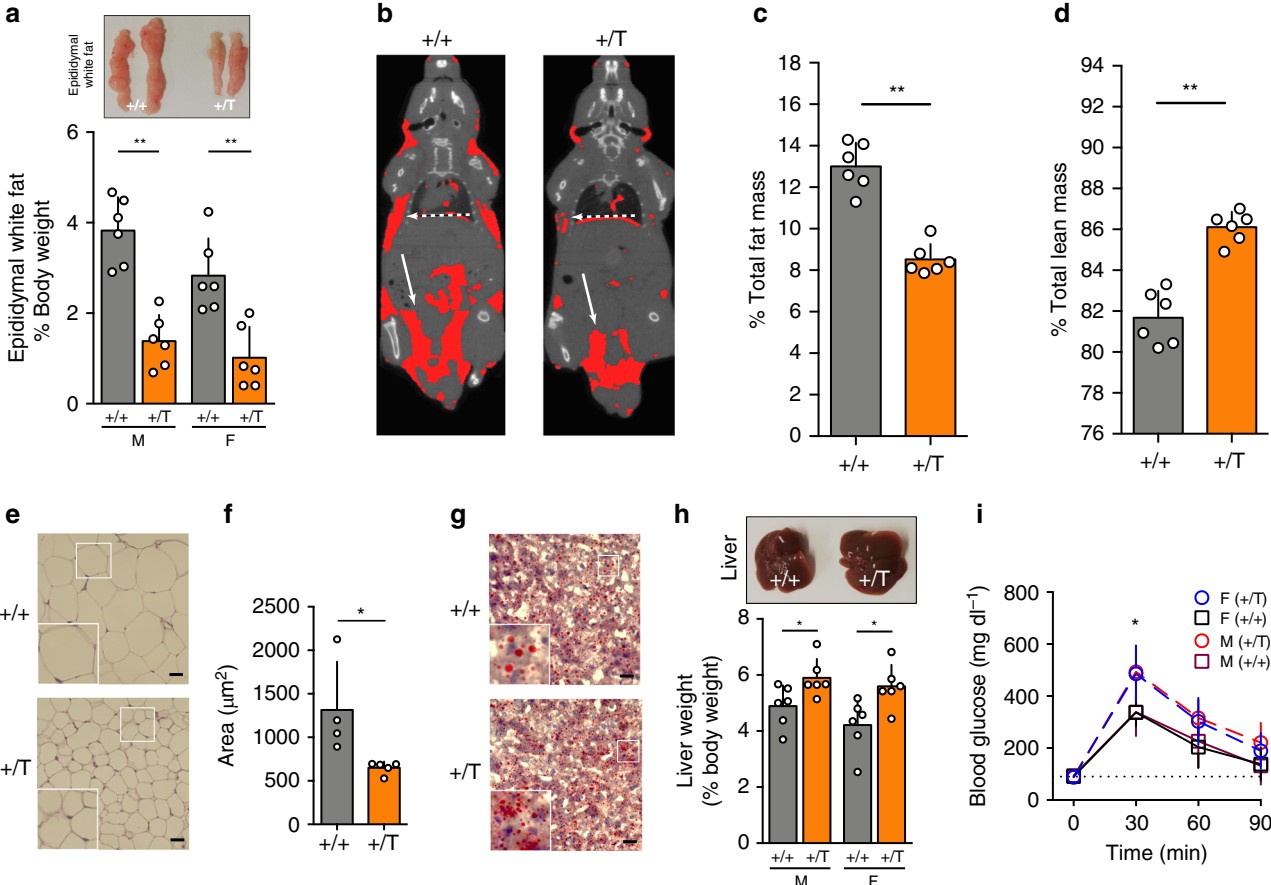

**Fig. 2 GH/IGF-1 hypersecretion leads to alterations in *Ghrhr*^Gpr101 mice body composition. a** Picture: Epididymal white fat from WT (+/+) and *Ghrhr*^Gpr101 (+/T). Bars: quantification of epididymal white fat weight normalized to total body weight ($n = 6$ mice/group, $p = 0.0022$ for males and females). **b** Representative CT images segmented for fat of 27-week-old WT (+/+) and *Ghrhr*^Gpr101 (+/T) mice ($n = 6$ mice per group). Total volume is in grayscale and fat volume is in red. Plain white arrows indicate subcutaneous fat and dashed arrows epididymal fat distribution. **c** Percentage of fat mass in WT (+/+) and *Ghrhr*^Gpr101 (+/T) mice ($n = 6$ mice per group, $p = 0.0022$) determined by CT-scan analysis. **d** Percentage of lean mass in WT (+/+) and *Ghrhr*^Gpr101 (+/T) mice ($n = 6$ mice per group, $p = 0.0022$) determined by CT-scan analysis. **e** 27-week-old WT (+/+) and *Ghrhr*^Gpr101 (+/T) representative histological sections of epididymal adipose tissue stained with H&E. Scale bar: 30 μm. This experiment was repeated at least three times. **f** Mean adipocyte area, quantified using at least four fields per whole-slide image, from at least four animals per group ($p = 0.0159$). **g** Representative liver histological sections from 27-week-old WT (+/+) and *Ghrhr*^Gpr101 (+/T) mice stained with ORO ($n = 4$ mice per group). Scale bars: 30 μm. **h** Picture: extracted livers from WT (+/+) and *Ghrhr*^Gpr101 (+/T). Bars: quantification of liver weight normalized to body weight ($n = 6$ mice per group, males: $p = 0.0303$, females: $p = 0.0130$). **i** GTT of 11 month-old WT (+/+) and *Ghrhr*^Gpr101 (+/T) mice. GTT was performed after 12 h of fasting. Glucose was injected IP to starved mice and blood was collected at indicated time points (0, 30, 60, and 90 min) to measure blood glucose levels (males: $n = 6$ mice per group, females: $n = 5$ mice per group). 0 min: $p = 0.5173$ for males and $p = 0.7302$ for females. 30 min: $p = 0.0260$ for males and $p = 0.0317$ for females. 60 min: $p = 0.1970$ for males and $p = 0.0952$ for females. 90 min: $p = 0.1320$ for males and $p = 0.4206$ for females. All data are presented as Mean ± S.D. For statistical analysis of all data, a two-sided Mann–Whitney test was used. ns not significantly different; *$p < 0.05$; **$p < 0.01$; ***$p < 0.001$. F Female, M Male.

Furthermore, we noted that the decreased fat mass in the *Ghrhr*^Gpr101 mice occurred due to significantly reduced adipocyte fat content and decreased mean adipocyte area (Fig. 2e, f). Lipid storage in the *Ghrhr*^Gpr101 mice was also reduced at the hepatic level (Fig. 2g). We also noted hepatomegaly in the *Ghrhr*^Gpr101 male and female mice, which may have occurred as a consequence of GH/IGF-1 induced organomegaly (Fig. 2h). Chronic GH hypersecretion in humans is associated with altered carbohydrate metabolism[20]. Such effects occurred in *Ghrhr*^Gpr101 mice of both sexes, with increased blood glucose concentrations following intraperitoneal (IP) administration of a glucose load confirming the diabetogenic action of GH in this model (Fig. 2i)[21]. The described phenotypic traits were recapitulated in another Tg line that incorporated fewer copies of FLAG-Gpr101 the transgene (Supplementary Fig. 3).

Next, we investigated the pituitary glands of the *Ghrhr*^Gpr101 mice to assess whether the source of excess GH and PRL secretion was due to abnormal somatotrope cell proliferation. Remarkably, gross pathological and microscopic examination of pituitaries from *Ghrhr*^Gpr101 showed no evidence of pituitary adenoma (Fig. 3a, b). There was no evidence of increased proliferation within the anterior pituitary as assessed by Ki-67 staining and hyperplasia was not present on reticulin staining (Fig. 3c, e). In addition, we excluded that the few Ki-67 cells were somatotropes by co-staining this proliferation marker with GH in immuno-fluorescence experiments (Fig. 3d). To better understand the cause of GH hypersecretion, we determined the extent of hormone presence in the anterior pituitary both at transcriptional and protein levels, both of which were elevated in the *Ghrhr*^Gpr101 animals (Fig. 3f, g). In parallel, we verified by PCR the amount of

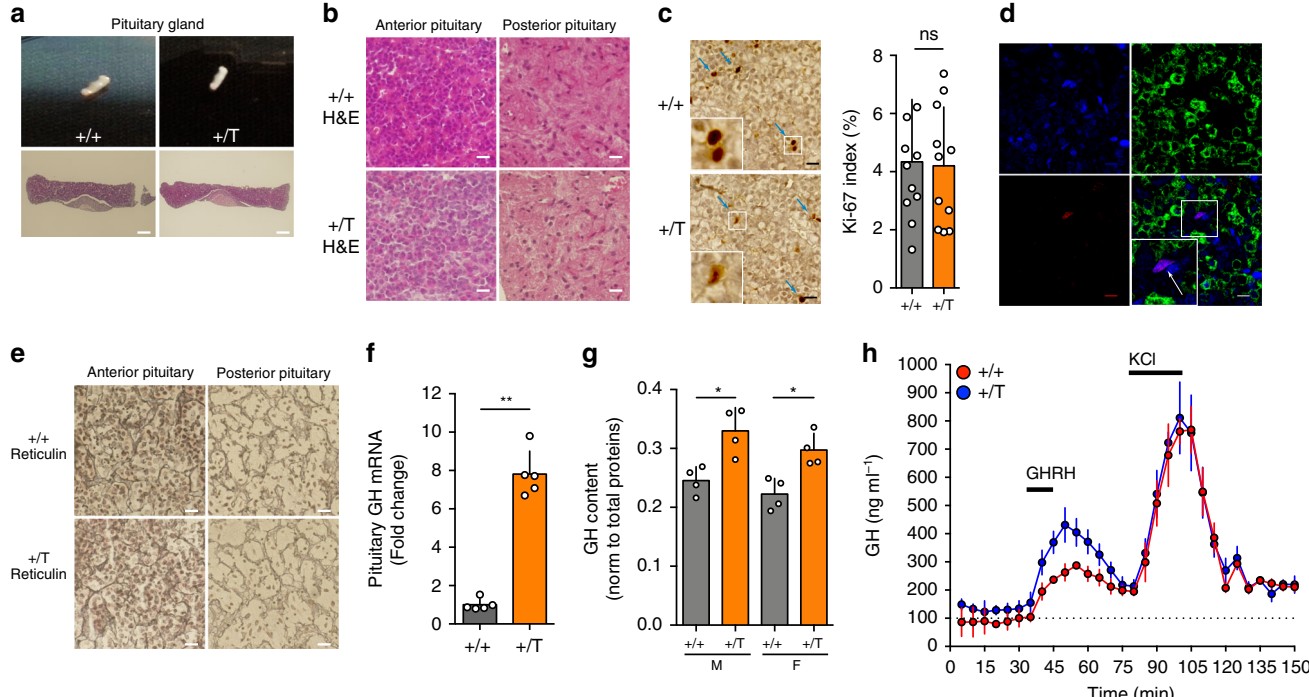

**Fig. 3 Gpr101 overexpression potentiates GH release but does not lead to hyperplasia/tumorigenesis. a** Upper panel: Macroscopic analysis of the pituitary gland from 27-week-old WT (+/+) and $Ghrhr^{Gpr101}$ (+/T) mice. Lower panel: microscopic visualization of pituitary sections after H&E staining. Scale bar: 150 µm. **b** High magnification of anterior and posterior pituitaries stained with H&E. Scale bar: 15 µm. **c** Left panel: Immunohistochemical staining of the anterior pituitary sections with the cell proliferation marker Ki-67 (scale bar: 15 µm). Blue arrows indicate Ki-67-positive nuclei staining. Right panel: quantification of the Ki-67 labeling index in pituitary sections of 27-week-old WT (+/+) and $Ghrhr^{Gpr101}$ (+/T) mice. The Ki-67 labeling index represents the percentage of positive nuclei stained by anti-Ki-67 antibody. $n = 11$ independent areas from staining section of WT (+/+) and $Ghrhr^{Gpr101}$ (+/T) ($n = 4$ mice per group, $p = 0.9487$). **d** Immunofluorescent staining of GH (green) and Ki-67 (Red). Scale bar: 10 µm. **e** Reticulin staining of the anterior and posterior pituitaries of WT (+/+) and $Ghrhr^{Gpr101}$ (+/T) mice (scale bar: 15 µm). **f** The expression of GH in the pituitary of WT and $Ghrhr^{Gpr101}$ mice (aged 27 weeks $n = 5$, $p = 0.0079$) was quantified by RT-qPCR. GAPDH was used as a control housekeeping gene. **g** The content of the GH protein was quantified by ELISA and normalized to total protein in pituitary lysates of both males and females of the WT (+/+) and $Ghrhr^{Gpr101}$ (+/T) genotypes (aged 29 weeks, $n = 4$, $p = 0.0286$). **h** Ex vivo pituitary superfusion analysis. Pituitary glands of WT (+/+) and $Ghrhr^{Gpr101}$ (+/T) (aged 29 weeks, $n = 3$ mice) were superfused at 0.1 ml min$^{-1}$ in superfusion chambers. Effluents were collected every 5 min for GH measurement. GHRH (100 nM) was added to the medium for 15 min and KCl (0.03 M) for 20 min (as it is indicated with arrows). GH secretion was quantified by ELISA at indicated time points. All the experiments were independently repeated three times unless stated otherwise. F Female, M Male. For statistical analysis of all data, a two-sided Mann–Whitney test was used unless stated otherwise. ns not significantly different; $*p < 0.05$; $**p < 0.01$; $***p < 0.001$.

mRNA for GHRH in the hypothalamus as well as Pit-1 and GHRHR in the pituitary and showed no differences between $Ghrhr^{Gpr101}$ and WT animals (Supplementary Fig. 4a–c). We confirmed that circulating levels of GHRH or SST were not significantly altered in $Ghrhr^{Gpr101}$ mice (Supplementary Fig. 4d, e). We then measured the response of somatotropes from $Ghrhr^{Gpr101}$ mice to the major canonical stimulatory signal, GHRH, in freshly extracted pituitaries using superfusion chambers. While basal GH levels were slightly elevated as compared with WT, the peak GH secretion in response to GHRH was significantly elevated in the $Ghrhr^{Gpr101}$ animals (Fig. 3h). Taken together, the evidence suggests that the chronic hormonal hypersecretion, skeletal overgrowth and altered body composition in $Ghrhr^{Gpr101}$ mice is most probably due to the direct action of Gpr101 on somatotropes to promote GH synthesis and release.

**GPR101 is constitutively coupled to $G_s$, $G_{q/11}$, and $G_{12/13}$.** Given the pronounced enhancement of hormonal secretion due to increased Gpr101 in the Tg animals, we undertook studies to clarify the signaling characteristics and pathways that underlie its actions. Previous work suggested that GPR101 acted through $G_s$ and increased cyclic adenosine monophosphate (cAMP)

levels[7,12,13]. We first established a GPR101-transfected model in Human Embryonic Kidney (HEK)-293 cells and detected a robust constitutive increase in cAMP levels using a GloSensor cAMP assay (Fig. 4a)[22,23]. In order to firmly establish the link between cAMP production and $G_s$, we used CRISPR/Cas9 genome editing to deplete the α subunit of the G-protein families in HEK293 cells (HEK293.ΔG$_s$, HEK293.ΔG$_{q/11}$, and HEK293.ΔG$_{12/13}$)[24,25]. We also used a cell line which was depleted of all G protein α subunits ($G_{s/olf}$, $G_{q/11}$ and $G_{12/13}$) except for the $G_{i/o}$ family (HEK293.ΔG$_{tot}$)[26]. As a consequence, the effect of GPR101 on cAMP in HEK293.ΔG$_s$ and HEK293.ΔG$_{tot}$ was abolished while being unaffected in cells lacking $G_{q/11}$ or $G_{12/13}$ (Fig. 4a, Supplementary Fig. 5a). A combined depletion of $G_{q/11}$ and $G_{12/13}$ with a siRNA approach did not affect the cAMP increase either (Supplementary Fig. 5b). Consistently, we found that gpr101, the murine ortholog (70.7% protein sequence identity to the human protein), induced a similar increase in basal cAMP levels in HEK293 cells (Fig. 4b). The cAMP levels were of a similar magnitude as those of a control orphan receptor that is constitutively coupled to $G_s$, GPR3 (Fig. 4b).

Next, we examined whether GPR101 could be coupled to other G proteins. To study $G_{q/11}$ activation, we measured inositol monophosphate (IP$_1$) accumulation, the stable downstream metabolite of IP$_3$

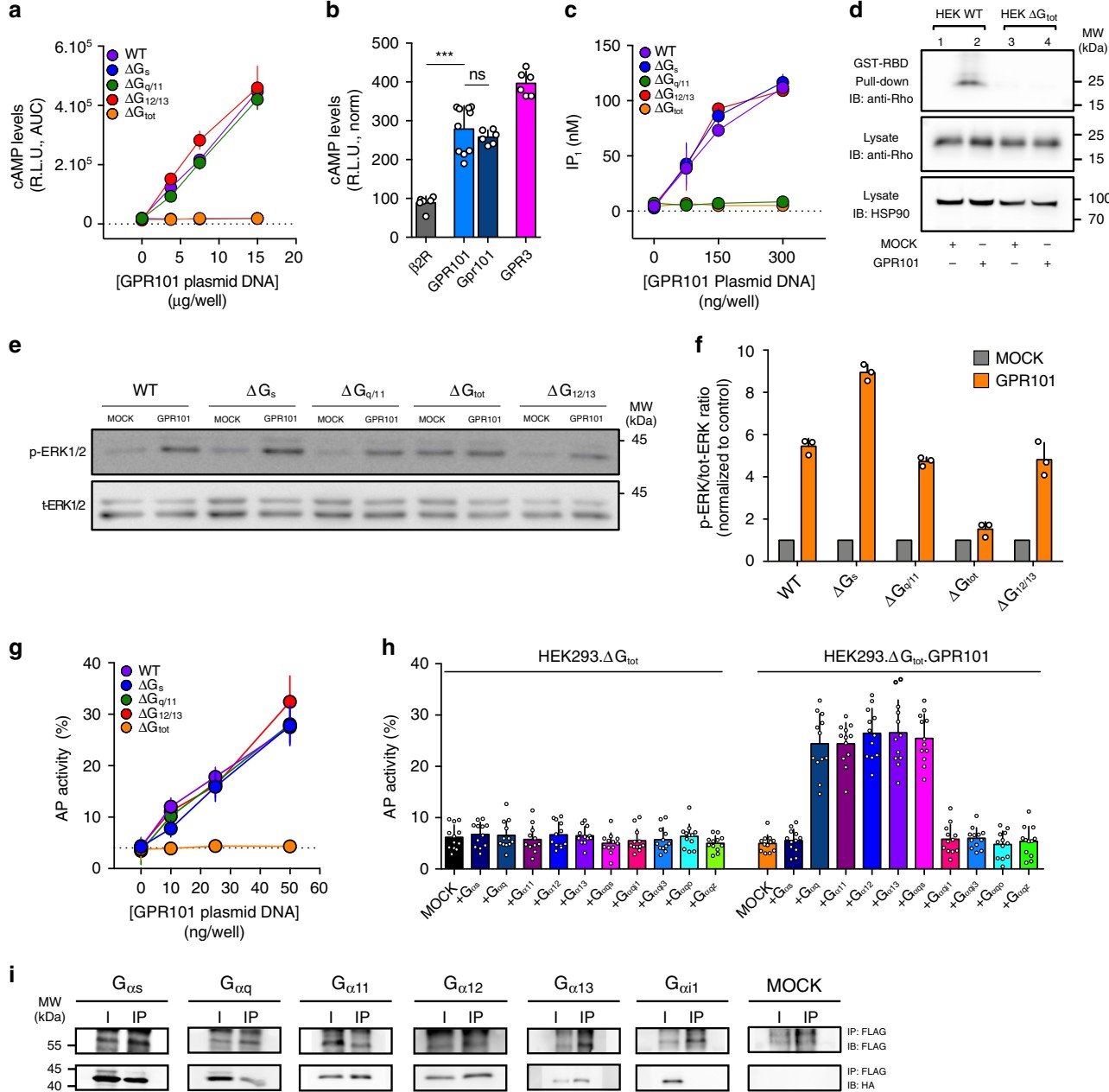

**Fig. 4 GPR101 is constitutively coupled to $G_s$, $G_{q/11}$, and $G_{12/13}$. a** Measurement of cAMP levels after GPR101 transient transfection in HEK293 WT, $\Delta G_s$, $\Delta G_{q/11}$, $\Delta G_{12/13}$, or $\Delta G_{tot}$. $n = 12$ independent experiments. **b** Comparison of constitutive cAMP levels obtained after transient transfection of pGlo.HEK293 with the indicated receptors: ß2AR ($n = 6$ independent experiments), GPR101 ($n = 10$ independent experiments), Gpr101 ($n = 6$ independent experiments), and GPR3 ($n = 6$ independent experiments). The values have been normalized to receptor expression to enable direct comparison. GPR101 vs ß2AR: $p = 0.0002$; GPR101 vs Gpr101: $p = 0.5622$. **c** Measurement of $IP_1$ levels after transient GPR101 transfection in HEK293 WT, $\Delta G_s$, $\Delta G_{q/11}$, $\Delta G_{12/13}$, or $\Delta G_{tot}$. $n = 4$ independent experiments. **d** Activated Rho was detected in lysates of HEK293 WT or HEK293$\Delta G_{tot}$ transiently transfected with GPR101 following precipitation with GST-Rho-binding domain (RBD). Shown are representative of at least three independent experiments. See text for details. **e** Shown are pictures of immunoblots for the determination of $ERK_{1/2}$ phosphorylation in WT HEK293 or HEK293 cells deficient for the indicated G proteins and transiently transfected with GPR101 or empty vector (MOCK). **f** Immunoblots were quantified by densitometric analysis. The p-$ERK_{1/2}$ to total $ERK_{1/2}$ ratio has been normalized to the MOCK condition. $n = 3$ independent experiments. **g** TGF α Shedding assay performed on HEK293 WT, $\Delta G_s$, $\Delta G_{q/11}$, $\Delta G_{12/13}$, or $\Delta G_{tot}$ transiently transfected with GPR101. Results are expressed as the percentage of AP activity in the conditioned medium. $n = 12$ independent experiments. **h** TGF α shedding assay in HEK293 $\Delta G_{tot}$ transiently transfected with empty vector (MOCK) or GPR101 alone or together with various $G_\alpha$ proteins and chimeric $G_\alpha$ proteins. Results are expressed as the percentage of AP activity in the conditioned medium. $n = 12$ independent experiments. **i** Co-Immunoprecipitation of FLAG-GPR101 with Anti-FLAG beads followed by immunodetection of HA-tagged $G_\alpha$ proteins with anti-HA antibody on WB membranes. Full scans of blots from **d**, **e**, and **i** can be found in the Source Data File. All data are Mean ± S.D. AUC area under curve, HSP90 heat shock protein 90. IB antibody used for blotted membrane, I input, IP immunoprecipitated fraction. Shown are representative pictures of three independent experiments. R.L.U. Relative Luminescence Unit. For statistical analysis of all data, a two-sided Mann–Whitney test was used. ns not significantly different; *$p < 0.05$; **$p < 0.01$; ***$p < 0.001$.

formed as a result of phospholipase Cβ (PLCβ) activation[27]. Increasing expression of GPR101 increased $IP_1$ concentrations in HEK293 cells; this effect was abolished in a $G_{q/11}$-null background but was maintained in the absence of other G proteins (Fig. 4c, Supplementary Fig. 5c, d). Next, we assessed $G_{12/13}$ coupling with a Rho pull-down activation assay and detected the activation of this pathway in the presence of GPR101 (Fig. 4d). We were unable to detect Rho activation in the absence of $G_\alpha$ proteins (except $G_{\alpha i/o}$), which pointed to the ability for GPR101 to couple to $G_{12/13}$ (Fig. 4d).

It is generally accepted that signaling pathways activated by different G proteins converge on the activation of the mitogen-activated protein kinase (MAPK) cascade. Thus, we monitored the phosphorylation of extracellular signal-regulated kinase 1/2 ($ERK_{1/2}$) in our different HEK293 cell lines. In unmodified cells, the presence of GPR101 spontaneously increased basal phospho-$ERK_{1/2}$ (p-$ERK_{1/2}$; Fig. 4e, f). The signal was abolished in HEK293.$\Delta G_{tot}$ while we observed that the activation of the pathway in HEK293.$\Delta G_{q/11}$ or HEK293.$\Delta G_{12/13}$ was unaffected (Fig. 4e, f). Interestingly, we noticed an increased signal for p-ERK in HEK293.$\Delta G_s$ compared to parental cell lines (Fig. 4e, f). The basal $ERK_{1/2}$ phosphorylation was increased in HEK293.$\Delta G_{tot}$ (Fig. 4e) but this was likely due to an elevated basal activity of the remaining $G_{i/o}$ in those cells as it disappeared following pertussis toxin (PTX) treatment (Supplementary Fig. 5f). Next, we used an alkaline phosphatase-transforming growth factor alpha (AP-TGF α) shedding assay that is able to detect both $G_{q/11}$ and $G_{12/13}$ downstream pathways[28,29]. Upon transfection of increasing amounts of GPR101, we observed a higher level of AP-TGF α activity in the medium as compared to a MOCK-transfected control (Fig. 4g, Supplementary Fig. 5e). In G-protein-depleted HEK293 cell lines devoid of either $G_{\alpha 12/13}$ or $G_{\alpha q/11}$ subunits, there was no impact of deletion of these $G_\alpha$ proteins on AP-TGF α release (Fig. 4g), which supports a combined $G_{q/11}$ and $G_{12/13}$ coupling for GPR101.

To further analyze the complex coupling profile of GPR101, we performed a rescue experiment with the shedding assay and independently transfected each native $G_\alpha$ protein with or without GPR101 in HEK293.$\Delta G_{tot}$ (Fig. 4h). As expected, the transfection of empty vector (MOCK) or $G_{\alpha s}$ together with GPR101 gave no signal over the background (Fig. 4h). The transfection of either $G_{\alpha q}$, $G_{\alpha 11}$, $G_{\alpha 12}$, or $G_{\alpha 13}$ resulted in enhanced shedding of AP-TGF α into the supernatant (Fig. 4h). The use of a chimeric promiscuous $G_\alpha$ proteins ($G_{\alpha qs}$) confirmed the $G_s$ coupling of the receptor and the absence of $G_{i/o}$ family ($G_{qi1}$, $G_{qi3}$, $G_{qo}$, and $G_{qz}$) activation (Fig. 4h). Finally, we confirmed biochemically the GPR101 coupling profile with a co-immunoprecipitation assay. We were able to detect the precipitation with GPR101 of $G_{\alpha s}$, $G_{\alpha q}$, $G_{\alpha 11}$, $G_{\alpha 12}$, and $G_{\alpha 13}$ but not $G_{\alpha i1}$ (Fig. 4i).

**GPR101 promotes GH secretion through $G_s$ and $G_{q/11}$.** In order to investigate the functional consequences of GPR101-dependent activation of various G proteins, we studied the rat somato-mammotrope cell line, GH3, that secretes GH and PRL[30]. Consistent with the HEK293 data, upon transfection with GPR101 we observed a significantly elevated basal level of cAMP and $IP_1$ (Fig. 5a, b). Furthermore, we confirmed that the presence of GPR101 resulted in a time-dependent increase in GH levels in the cell culture supernatant (Fig. 5c). Next, we identified the pathways involved in the GPR101-mediated increase in GH release by depleting $G_{\alpha s}$, $G_{\alpha q/11}$, or $G_{\alpha 12/13}$ in GPR101-transfected GH3 cells using an siRNA-based approach (Fig. 5d). The ability of siRNAs to blunt $G_s$- and $G_{q/11}$- mediated GH release was validated by using GHRHR and the Ghrelin (GHS) receptor (GHSR) as controls, respectively (Fig. 5d)[31,32]. In GPR101-transfected cells,

depletion of either $G_{\alpha s}$ or $G_{\alpha q/11}$ drastically reduced GH concentration, while the knockdown of $G_{\alpha 12/13}$ had no significant effect (Fig. 5d).

Next, we sought to better characterize the downstream events linking GPR101, $G_s$, $G_{q/11}$ and the secretion of GH. In somatotropes, PKA and PKC are downstream effectors for $G_s$-AC-cAMP and $G_{q/11}$-PLC β, respectively[33]. We assessed the effect of inhibiting PKA and PKC with H89 and Calphostin, respectively, on GPR101-mediated GH increases[34]. Both inhibitors strongly impaired GPR101-mediated GH secretion (Fig. 5e). To determine if this was due to reduced GH synthesis, GH exocytosis, or both, we measured GH mRNA in our cell cultures as a surrogate for GH production. The increased GH mRNA seen in cells transfected with GPR101 was reduced by the PKA inhibitor H89 but not by the PKC inhibitor Calphostin (Fig. 5f, g).

We confirmed by Western Blot the direct stimulation of PKA and PKC by GPR101 (Fig. 5h) and we found with siRNA depletion of $G_\alpha$ subunits that only $G_s$ contributed to PKA activation while both $G_s$ and $G_{q/11}$ played a role in PKC activation (Fig. 5i, j). The inhibition of both pathways completely blunted the PKC activation while $G_{\alpha 12/13}$ depletion had no effect (Fig. 5j).

**GPR101 has no effects on proliferation.** In the $Ghrhr^{Gpr101}$ mice we had noted a large increase in GH secretion, but no somatotrope proliferation or tumorigenesis. We therefore studied the potential links between $G_s$ and $G_{q/11}$ pathway activation and proliferation in GH3 cells. Transfection of increasing amounts of GPR101 had a negligible impact on proliferation, as compared with the marked proliferation induced by GHRH stimulation of GHRHR (Fig. 6a, b). We hypothesized that the cAMP increase induced by GPR101 might be insufficient to trigger proliferation. However, although co-transfection of GHRHR (activated by GHRH) and GPR101 leads to increased cAMP production over and above that obtained with GHRH-activated GHRHR (Fig. 6c), this did not translate into increased proliferation (Fig. 6b). In order to fully exclude a concentration-dependent effect, we repeated these experiments with a full concentration range of GHRH in the presence of increasing amounts of GPR101 (Fig. 6d, e). The potentiation of the cAMP increase was confirmed in GH3 cells, with a leftward displacement of the concentration-response curve, with no modification of GHRH maximal efficacy ($E_{max}$) (Fig. 6d). Again, the observed increase of GHRH half maximal effective concentration ($EC_{50}$) on cAMP increase did not translate into an increased potency for GHRH-induced proliferation (Fig. 6e). This likely suggests that the cAMP generated by GPR101 does not lead to a stimulation of cellular proliferation. This unexpected divergent effect between the effect of GPR101 and GHRHR signaling on proliferation was further explored in siRNA studies of G-protein subfamily depletion (with the GHSR receptor used as a positive control for constitutive $G_{q/11}$ and $G_{12/13}$ activation) (Fig. 6f). These studies confirmed in our system the established roles in somatotrope proliferation of GHRHR (through $G_s$) and of GHSR (through $G_{q/11}$ and $G_{12/13}$)[35,36]. For GPR101, the siRNA directed against $G_{\alpha s}$ resulted in an unexpected increase in proliferation, which echoed the finding of increased phospho-ERK activity following $G_{\alpha s}$ depletion described above (Fig. 4e, f). This paradoxical increase of proliferation was also present when $G_{\alpha s}$ was depleted in the presence of both GHRHR and GPR101 (Fig. 6f). We reasoned that in GPR101-transfected GH3 cells, the downstream activation of $G_s$-AC-cAMP could mitigate the proliferative effect of the activation of $G_{q/11}$-PLC β PKC. In keeping with the siRNA experiments, in cells transfected with GPR101, the direct adenylate cyclase activator forskolin (FSK) and PKA activator 8-Br-cAMP had negative impacts on proliferation while the PKA inhibitor H89 promoted

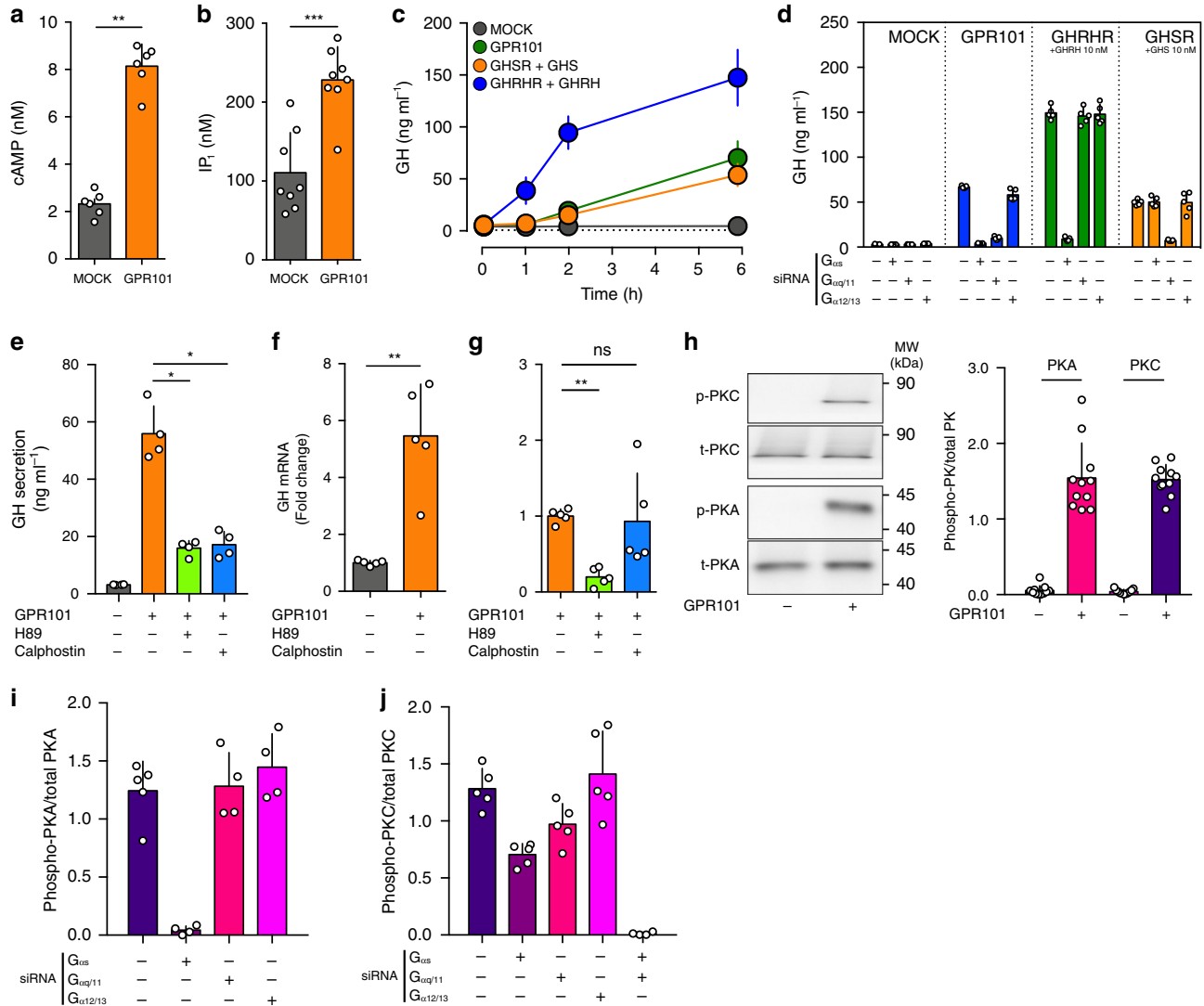

**Fig. 5 GPR101 promotes GH secretion through $G_s$ and $G_{q/11}$.** All the experiments presented here were performed on the GH3 pituitary cell line. **a** Determination of cAMP levels (by ELISA) following transient transfection with MOCK or GPR101 plasmid ($p = 0.0022$). **b** Determination of $IP_1$ levels (by ELISA) following transient transfection with MOCK or GPR101 plasmid ($p = 0.0006$). **c** Time-dependent (0, 1, 2, and 6H) measurement (by ELISA) of GH secretion in the cell culture supernatant. The cells were transfected with MOCK (dark grey), GPR101 (green), GHSR (orange), or GHRHR (blue) for 24 h, then starved for 3 h. For GHSR and GHRHR, cells were stimulated with their respective ligands (GHS or GHRH, 10 nM). GPR101 and MOCK received a vehicle treatment as control. **d** GH determination in the cell culture supernatant after 24h-treatment with various siRNAs ($G_{αs}$, $G_{αq/11}$, or $G_{α12/13}$), 24h-transfection with expression plasmids containing receptors (MOCK, GPR101, GHSR, or GHRHR), 3h-starvation and 6h-stimulation with indicated agonists (GHS or GHRH, 10 nM). **e** GH secretion was determined (by ELISA) in the cell culture supernatant following transfection with GPR101 (or MOCK) and treated with vehicle, H89 (10 μM, $p = 0.0286$) or Calphostin (10 μM, $p = 0.0286$). **f, g** Rat GH mRNA determination by RT-qPCR following transfection with MOCK or GPR101 ($p = 0.0079$) (**f**) and treatment with PKA & PKC inhibitors H89 (10 μM, $p = 0.0079$) and Calphostin (10 μM, $p = 0.6905$) (**g**), respectively. **h** Left: Immunoblot for the detection of phosphorylated PKA and PKC in GH3 cells following transfection with MOCK or GPR101. Right: Quantification by densitometry of immunoblots. Normalization was performed compared to total PKA and PKC proteins in cell lysate. Full scans of blots are available in the Source Data file. **i, j** Quantification by densitometry of immunoblots for PKA and PKC in the presence of GPR101 and different siRNAs. The antiphosphorylated antibody has been normalized to the signal from the antibody against total protein. All data are Mean ± S.D. of $n = 8$ (**b**), $n = 6$ (**a, c**), $n = 5$ (**d, f, g, j**), $n = 4$ (**e, i**), and $n = 3$ (**h**) independent experiments. For statistical analysis of all data, a two-sided Mann–Whitney test was used. ns not significantly different; *$p < 0.05$; **$p < 0.01$; ***$p < 0.001$.

proliferation (Fig. 6g). Thus, in GH3 cells, the activation of $G_s$ by GPR101 appears to prevent proliferation induced by $G_{q/11}/G_{12/13}$, while promoting GH secretion.

**PKC activation in somatotropes is a GPR101 signature in vivo.**
Collectively, our results in GH3 cells pointed to PKC as a characteristic feature of GPR101 downstream activation due to the

additive effects of the $G_s$-AC-cAMP and $G_{q/11}$-PLC β- Diacylglycerol (DAG) activation. In Tg $Ghrhr^{Gpr101}$ mice, we measured cAMP and $IP_1$ levels from pituitaries and confirmed that they were significantly elevated as compared to WT mice (Fig. 7a, b). We then used an antibody against activated phospho-PKC (at Threonine 638) to demonstrate the presence of significant pools of phospho-PKC in somatotropes from $Ghrhr^{Gpr101}$ animals, whereas phospho-PKC activity was low in WT control pituitaries (Fig. 7c).

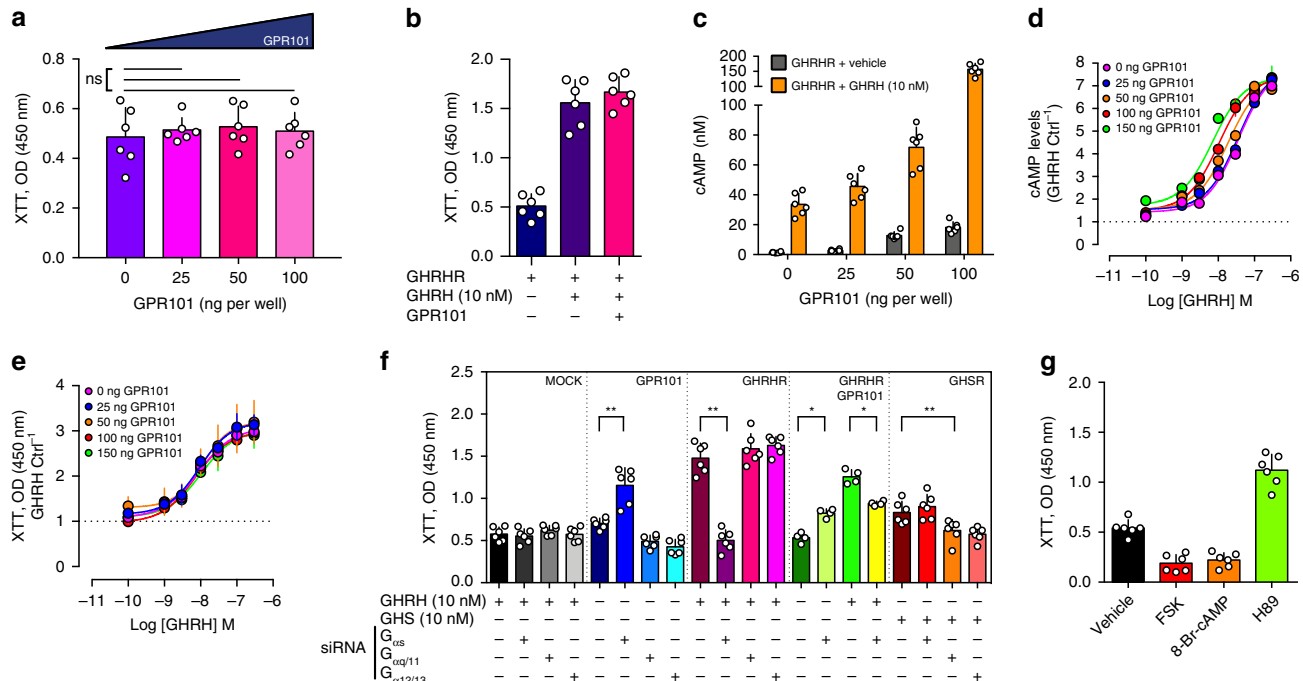

**Fig. 6 GPR101 does not increase GH3 cell proliferation. a** Proliferation was measured with XTT cell proliferation kit on GH3 cells transiently transfected with increasing amounts (0, 25, 50, and 100 ng) of pcDNA3.1 FLAG-GPR101 plasmid (0 ng vs 25 ng $p = 0.8182$; vs 50 ng $p = 0.5887$; vs 100 ng $p = 0.6991$). **b** Quantification of proliferation (by using the XTT reagent) of GH3 cells co-transfected with GHRHR and GPR101 (or MOCK) and treated with vehicle or GHRH at final concentration of 10 nM. **c** Comparison of cAMP levels (measured by ELISA) in GH3 cells transfected with GHRHR (100 ng) in the presence of increasing amounts (0, 25, 50, and 100 ng) of GPR101 plasmids, and treated with vehicle (dark grey) or GHRH (10 nM, orange). **d**, **e** GH3 cells transfected with GHRHR (100 ng) and increasing amounts of FLAG-GPR101. The cells were treated with increasing concentration of GHRH. **d** cAMP levels measured by ELISA and normalized to vehicle condition. **e** Proliferation measured with XTT assay. **f** Effect of the siRNA-mediated depletion of different G protein α subunits (G$_{αs}$, G$_{αq/11}$, or G$_{α12/13}$) on GH3 proliferation measured Briefly, GH3 cells were incubated with siRNAs (G$_{αs}$, G$_{αq/11}$, or G$_{α12/13}$, at a final concentration of 1 μM) for 24 h, and then transfected with MOCK, GPR101 (NTS vs G$_{αs}$ $p = 0.0022$), GHRHR (NTS vs G$_{αs}$ $p = 0.0022$), GPR101+GHRHR (untreated NTS vs G$_{αs}$ $p = 0.0286$, GHRH-treated NTS vs G$_{αs}$ $p = 0.0286$) or GHSR (GHS-treated NTS vs G$_{αq/11}$ $p = 0.0065$). GHRHR- and GPR101 +GHRHR-transfected cells were stimulated with GHRH (10 nM) and GHSR-transfected cells with GHS (10 nM). **g** Determination of the proliferation of GPR101-transfected GH3 cells treated with vehicle or different pharmacological agents, such as FSK (adenylate cyclase activator, 10 μM), 8-Br-cAMP (PKA activator, 10 μM), and H89 (PKA inhibitor, 10 μM). All data are Mean ± S.D. of $n = 6$ independent experiments (except for the co-transfection of GHRHR with GPR101 in panel **f** and for experiments of panels **d**, **e** where $n = 4$). For statistical analysis of all data, a two-sided Mann–Whitney test was used. ns not significantly different; *$p < 0.05$; **$p < 0.01$; ***$p < 0.001$.

This led us to consider whether higher PKC activation could be detected in human pituitaries with high levels of GPR101 expression. Thus, we used pituitary adenomas from X-LAG patients that are known to overexpress GPR101[7]. We compared phospho-PKC signal intensity in Formalin-fixed paraffin-embedded (FFPE) tissue from pituitary adenomas in X-LAG patients ($n = 3$) as compared with a group of acromegaly patients without GPR101 duplications ($n = 9$). As shown in Fig. 7d, e, tumors from X-LAG syndrome cases exhibited markedly elevated phospho-PKC staining, as compared to the low level seen in acromegaly controls, including tumors from patients with *aryl hydrocarbon receptor interacting protein* (*AIP*) mutations, an aggressive, treatment-resistant genetic form of acrogigantism. These results indicate that this signaling pathway in somatotropes involving G$_s$, G$_{q/11}$, and G$_{12/13}$ linked to increased PKC activity in *Ghrhr^Gpr101* and GH3 cells is present in human pituitary tissues with high GPR101 expression levels.

## Discussion

The potential role of GPR101 in somatotropes initially came to light with the description of X-LAG, in which a duplication on chromosome Xq26.3 including *GPR101* is associated with infant-onset GH and PRL-secreting pituitary adenomas that express high levels of GPR101[7–9,37]. Subsequently, we demonstrated that

GPR101 expression levels appear to mirror the expansion of somatotrope populations in the maturing human fetal pituitary[13]. However, fundamental information on the consequences of GPR101 signaling in somatotropes has been missing.

Our results in HEK and GH3 cells and in *Ghrhr^Gpr101* mice provide important insights into the role of increased GPR101 expression and activity in pituitary somatotropes. The phenotype of the *Ghrhr^Gpr101* mice was one of chronically elevated GH and IGF-1 secretion, with the expected effects on growth and body composition[38,39]. In addition, hyperprolactinemia was seen, which indicates that GPR101 facilitates the enhanced secretion of both GH and PRL. Cell subpopulations that co-secrete various hormones, such as, somatomammotropes that co-secrete GH and PRL, could be the source of this hyperprolactinemia. Somato-mammotropes and other co-secreting pituitary cells are a well-described phenomenon in normal mice, rats, and humans and in somatotropinomas, including the rat GH3 cell line[40–42]. Notably, the presence of important pools of cells secreting both GH and PRL has been repeatedly confirmed by single cell transcriptomic analysis[43–45]. As in other settings, increased body size and bone growth (vertebrae and long bones) due to GH/IGF-1 elevation was accompanied by decreased fat mass and increased lean mass in the *Ghrhr^Gpr101* mice versus wild-type controls (See Fig. 2)[46]. These features demonstrate that increased expression of Gpr101

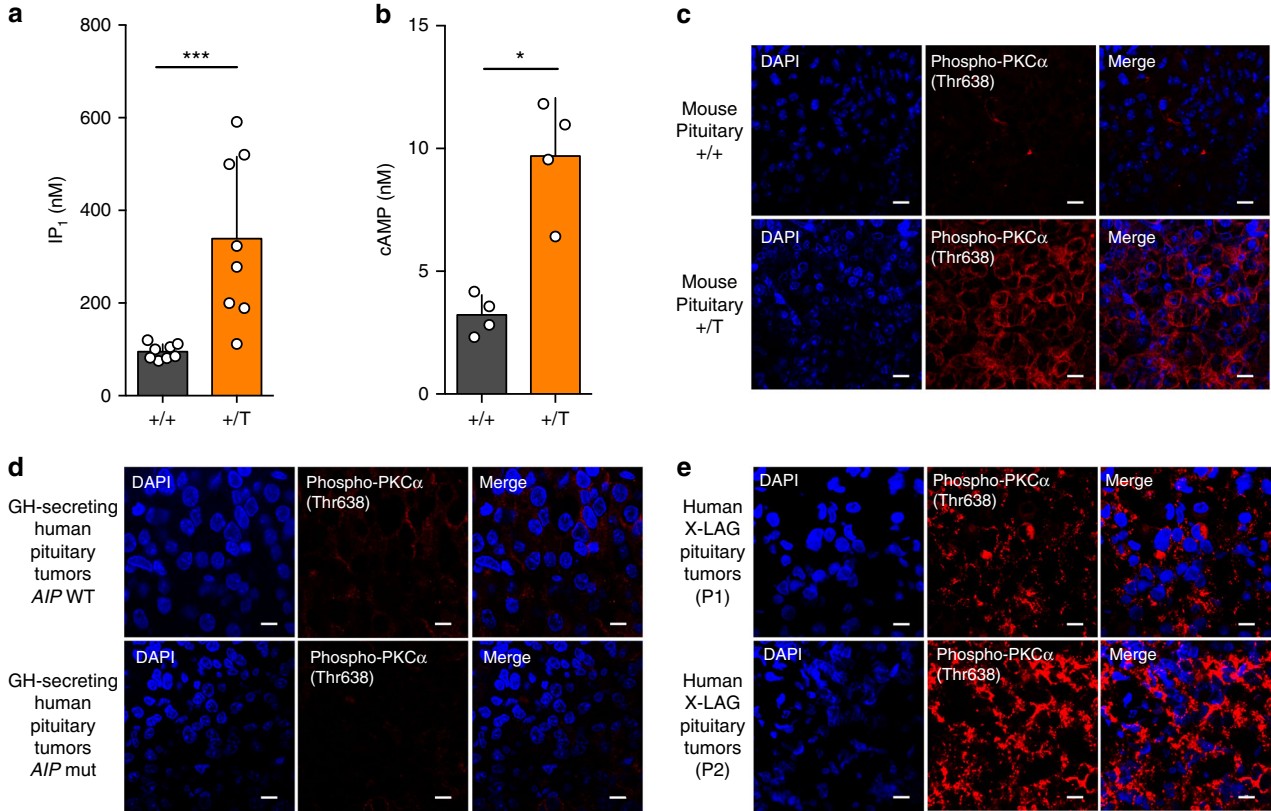

**Fig. 7 PKC activation is a signature in somatotropes with high GPR101 expression in vivo. a** Quantification of IP$_1$ levels in WT ($+/+$) and *Ghrhr*$^{Gpr101}$ ($+/T$) pituitaries by ELISA in mice aged 29 weeks ($n = 8$ mice per group, $p = 0.0005$). **b** Quantification of cAMP levels in WT ($+/+$) and *Ghrhr*$^{Gpr101}$ ($+/T$) pituitaries by ELISA in mice aged 29 weeks ($n = 4$ mice per group, $p = 0.0286$). **c** Immunofluorescent staining of Phospho-PKCα (Thr638) (in red) and DAPI (in blue) in pituitaries from 29-week-old WT ($+/+$) and WT ($+/+$) mice ($n = 4$ mice/group). The results show that the staining for Phospho-PKCα (Thr638) is increased in tg *Ghrhr*$^{Gpr101}$ mice compared to WT mice. Scale bar: 10 μm. **d, e** Illustrative examples of immunofluorescent staining of Phospho-PKCα (Thr638) (in red) and DAPI (in blue) in human GH-secreting pituitary adenomas (**d**, in total $n = 9$ patients) that are either *AIP* WT (upper panel, $n = 6$) or *AIP* mutated (lower panel, $n = 3$) and illustrative examples of human X-LAG pituitary tumors from two different patients (**e**, $n = 3$ patients). The results show that Phospho-PKC (Thr638) staining in human X-LAG pituitary tumors is elevated. Photos were taken at ×60 magnification. Scale bar: 10 μm. All data are Mean ± S.D. For statistical analysis of all data, a two-sided Mann–Whitney test was used. *$p < 0.05$; **$p < 0.01$; ***$p < 0.001$.

in the mouse pituitary is sufficient to chronically disrupt the GH-IGF-1 axis and significantly alter body size, composition and metabolism. Further aspects of Gpr101-related hormonal secretion in mice remain to be explored, such as, the important issue of potential alterations in GH pulsatility. Similarly, the magnitude of the secretory responses to GHRH stimulation seen in our ex vivo experiments of pituitary tissue needs to be balanced against the greater magnitude of GH responses to GHRH that occur in vivo. The hyperprolactinemia that is also encountered in the *Ghrhr*$^{Gpr101}$ mice requires specific studies to determine the precise mechanisms by which PRL dysregulation occurs and how this impacts the phenotype of these animals.

It is long-established that both G$_q$- and G$_s$-mediated signaling can drive GH secretion through distinct pathways converging on [Ca$^{2+}$]$_i$ increases triggering vesicles exocytosis[38]. Here, we show that GPR101 can couple constitutively to multiple G-protein families, namely G$_s$, G$_{q/11}$, and G$_{12/13}$. This behavior is not uncommon among GPCRs, especially those from the Rhodopsin family[47] and has been observed for some pituitary receptors. For example, the GHSR has a similar complex coupling profile (G$_{q/11}$, G$_{i/o}$, and G$_{12/13}$) and high constitutive signaling activity[48,49]. Another notable example of such promiscuous coupling is the pituitary adenylate cyclase-activating polypeptide (PACAP) type I receptor that possesses some GH secretagogue capacity[50–52].

Constitutive activity of GPR101 is a notable feature that most likely plays a key role in the physiological functions of the receptor and the pathophysiology of X-LAG syndrome. This is mirrored by other diseases in endocrinology where the impact of constitutive activity has already been documented. For example, the GHSR constitutive activity, when impaired by mutations, is related to short stature[53]. Furthermore, activating mutations of *GNAS* affecting G$_{αs}$ lead to McCune-Albright syndrome (postzygotic mosaicism) and acromegaly (somatic mutations), while activating mutations of the thyrotropin-stimulating hormone (TSH) and luteinizing hormone (LH) receptor are associated with diseases such as hyperfunctioning thyroid adenoma and familial male precocious puberty, respectively[54,55].

The *Ghrhr*$^{Gpr101}$ mice developed chronic GH/PRL hypersecretion in the absence of adenoma or hyperplasia, indicating that Gpr101 overexpression can act as a powerful promoter of GH secretion in mice, even in a non-tumoral setting. In this aspect of tumorigenesis, the *Ghrhr*$^{Gpr101}$ mice diverge from the phenotype of X-LAG; this indicates that X-LAG is a multifactorial process and important factors in its pathogenesis are not addressed by the current models and remain to be explained. Our results suggest that increased GPR101 expression in our models intensify G$_s$ and G$_{q/11}$ pathway basal tone due to the constitutive receptor activity. Furthermore, G$_{q/11}$-PLC β-PKC mediated activation appears to be a prominent pathway by which GPR101 can modulate hormonal secretion in the models studied.

Abnormal or constitutive activation of the G$_s$-cAMP pathway is one of the best-established mechanisms for somatotropinoma

formation and GH hypersecretion in various forms of acromegaly[56]. As noted above, a strong link between $G_s$ and somatotrope tumorigenesis due to activating *GNAS* mutations has been established in Mc Cune-Albright syndrome and in up to 40% of sporadic cases of acromegaly[57]. Moreover, ectopic secretion of GHRH leads to pituitary hyperplasia and adenomas and it has been convincingly demonstrated that this was driven directly by the activation of $G_s$ through the GHRHR in pituitary somatotropes[58]. However, our results suggest that $G_s$ activity in somatotropes does not invariably lead to proliferation and, via GPR101, may even counteract it. Divergent functional effects between different cAMP-elevating receptors in specialized cells has been documented for several decades[59,60]. The results we obtained may be a manifestation of such compartmentalization of signaling but will require further investigation to be firmly demonstrated. Other possible explanations exist for the differences observed in cellular response to GHRHR or GPR101, like the pattern of stimulation triggered by the two receptors. Indeed, when it is expressed, GPR101 activates the $G_s$ continuously, in a chronic fashion, while GHRHR responds only to an acute stimulation by GHRH.

Collectively, our results demonstrate that GPR101 can drive GH secretion in the pituitary through a constitutive activation of both $G_s$ and $G_{q/11}$. We propose a model where the effects of GPR101 in somatotropes result in an elevated GH secretion in response to physiological stimuli and potentially in somatotrope axis dysfunction (Fig. 8).

## Methods

**Reagents**. All chemicals used were from Sigma–Aldrich (St. Louis, MO, USA) unless stated otherwise. Human Ghrelin (trifluoroacetate salt, Cat. No 4033077.0500) and human GHRH (acetate salt, Cat. No 4011472.0001) were purchased from Bachem (Switzerland). Mouse ghrh was from Phoenix Pharmaceuticals (Belmont, CA, USA, Cat. No 031-14). Calphostin C (PKC inhibitor, Cat. No C6303) and H89 (PKA inhibitor, Cat. No B1427) were purchased from Sigma–Aldrich. 8-Br-cAMP is from Merck Millipore (Burlington, Massachusetts, United States, Cat. No 37060590). FSK is from Gentaur (Kampenhout, Belgium, Cat. No 203-16384-84). The validated siRNAs and Accell Delivery Medium (Cat. No B-005000-100) were bought from Dharmacon (Horizon Discovery, Cambridge, UK). Primers used for cloning, RT-PCR and genotyping were from Integrated DNA Technologies (IDT, Leuven, Belgium).

**Plasmids**. Human GPR101, β2-adrenoceptor (β2AR) and GPR3 were amplified from genomic DNA extracted from HEK293 cells. Mouse gpr101 was amplified

from mouse tail genomic DNA. All receptors were cloned into the pIRESpuro vector (Clontech Laboratories, Mountain View, CA, USA) (for stable transfections) and/or pcDNA3.1 (Invitrogen Corporation, Carlsbad, CA, USA) (for transient transfections) after addition of the FLAG epitope (DYKDDDDK) at the N-terminus, preceded by the signal sequence (ss) KTIIALSYIFCLVFA[61], unless specified otherwise. Expression vectors encoding for the human GHRH and the GHS receptors were purchased from cDNA Resource Center (Bloomsburg, PA, USA, Cat. No GHRHR00000 and GHSR0A0000, respectively). The pGloSensorTM-22F cAMP (cAMP GloSensor™) plasmid was obtained from Promega Corporation (Madison, WI, USA). Plasmids for AP-TGFα shedding assay (pCAGGS/AP-TGFα) and the chimeric $G_{aqs}$, $G_{aqi1}$, $G_{aqi3}$, $G_{aqo}$, and $G_{aqz}$ proteins were described elsewhere[28]. The $G_{ai1}$, $G_{aq}$, $G_{a11}$, $G_{a12}$, and $G_{a13}$ proteins were amplified from the cDNA of HEK293 cell mRNA. $G_{as}$ was amplified from plasmid obtained at cDNA Resource Center (Bloomsburg, PA, USA). All G proteins were cloned into the pcDNA3.1 (Invitrogen, Carlsbad, CA, USA) after PCR addition of the HA (YPYDVPDYA) epitope at the N-terminus. The sequence of all plasmid constructs was validated by sanger sequencing (GIGA genomic platform, Liège, Belgium).

**Cell culture and transfection**. HEK293 cells (ATCC, Manassas, VA, USA) were grown in Dulbecco's Modified Eagle Medium (DMEM, Lonza, Verviers, Belgium) supplemented with 10% fetal bovine serum (FBS) (International Medical Products, Cat. No P40-37500, lot P160105), 1% penicillin/streptomycin (Lonza, Verviers, Belgium), and 2mM L-glutamine (Lonza, Verviers, Belgium) at 37 °C and 5% $CO_2$. Stable pGlo cell lines were selected for 5 weeks with hygromycin B Gold (160 μg ml$^{-1}$, InvivoGen). Resulting cell clones were checked by FACS analysis. CRISPR/Cas9 generated HEK293 depleted for G proteins have been described elsewhere (ΔG$_{s/olf}$[25], ΔG$_{q/11}$[62], ΔG$_{12/13}$[63], and ΔG$_{tot}$[26]). Forty-eight hour before the experiment, wild-type (WT) or CRISPR/Cas9 HEK293 cells were transfected with plasmids using calcium phosphate precipitation method, unless stated otherwise. Rat pituitary tumor GH3 cells (Sigma–Aldrich, Cat. No 87012603) were grown in Ham's F10 medium (Life technologies, Cat. No 31550023) supplemented with 15% horse serum (HS, Sigma–Aldrich, Cat. No H1138), 2.5% FBS, 1% penicillin/streptomycin, and 2 mM L-glutamine and maintained at 37 °C in an atmosphere of 95% air and 5% $CO_2$. Forty-eight hour before assay, GH3 cells were transfected with plasmids using lipofectamine™ 3000 transfection reagent (Life technologies, Cat. No L3000008), according to the manufacturer's recommendations.

**GloSensor cAMP assay**. We determined cAMP levels with the GloSensor™ technique[22] according to the manufacturer's instructions (Promega, Madison, WI, USA). For constitutive activity, HEK293 cells stably expressing the GloSensor Plasmid 22 F were transiently transfected with increasing concentrations (0, 7.5, 15, and 30 μg) of plasmids containing GPR101 or other receptors (β2AR, Gpr101 and GPR3). Forty-eight hour later, cells were detached and incubated 1 h in the dark at room temperature in HBSS assay buffer (120 mM NaCl, 5.4 mM KCl, 0.8 mM $MgSO_4$, 10 mM HEPES; pH 7.4, 10 mM glucose) containing IBMX (100 μM). Cells were distributed into 96-well plates (microplate, PS, 96-well, F-bottom, white, lumitrac, Greiner bio-one, Cat. No 655075). Following injection of the luciferase substrate, luminescence was directly recorded for 30 min on a Centro XS3 LB 960 reader (Berthold Technologies, Bad Wildbad, Germany). For the experiments on CRISPR/Cas9 HEK293 lines, ΔG$_{s/olf}$, ΔG$_{q/11}$, ΔG$_{12/13}$, ΔG$_{tot}$ and parental cells were

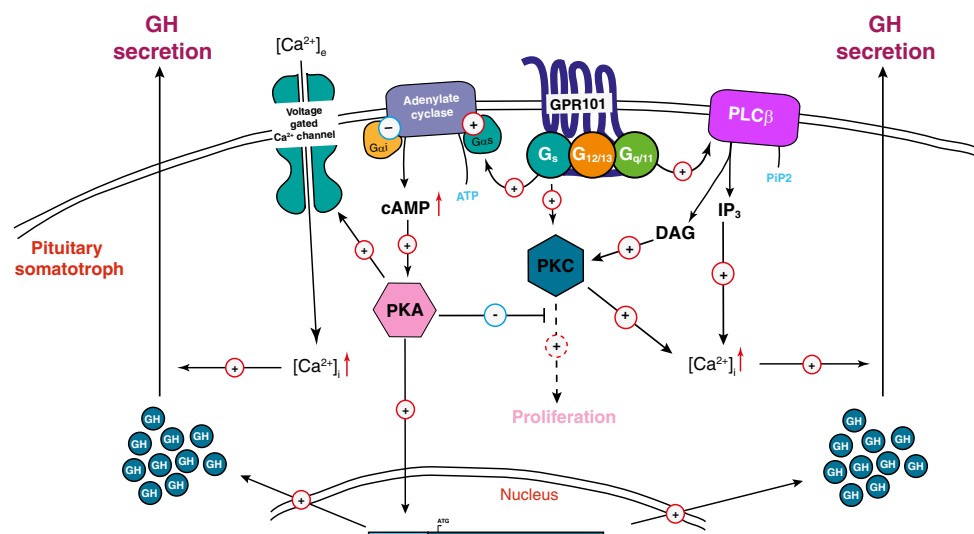

**Fig. 8 Proposed model for GPR101-induced GH secretion.** $[Ca^{2+}]_i$: Intracellular Calcium; cAMP 3',5'-cyclic Adenosine Monophosphate, DAG Diacylglycerol, GH Growth Hormone, $IP_3$ Inositol Triphosphate, $PIP_2$ Phosphatidylinositol 4,5-bisphosphate, PKA Protein Kinase A, PKC Protein Kinase C, PLCβ Phospholipase Cβ.

transiently co-transfected with GPR101 (or MOCK) and GloSensor cAMP biosensor and subjected to the same procedure.

**Measurement of cAMP levels by ELISA.** The Direct cAMP ELISA Kit (Enzo Life Sciences, Cat. No ADI-900-066) was used for the determination of cAMP levels in GH3 cells and mouse tissues. GH3 cells were grown in 24-well culture plates and transfected with increasing concentrations (0, 25, 50, 100, and 150 ng) of plasmid (MOCK or GPR101). After 48 h, the media was removed, the cells were incubated in HCl (0.1 M) for 10 min at room temperature and the cell lysate was centrifuged. For the experiments of co-transfection of GHRHR (50 ng) with increasing concentrations (0, 25, 50, 100, and 150 ng) of GPR101, cells were stimulated with vehicle (PBS) or GHRH (0–300 nM) for 15 min at 37 °C before cell lysate. Pituitary tissues from 29-week-old $Ghrhr^{Gpr101}$ and WT were collected and flash frozen in liquid nitrogen. Then, they were homogenized in HCl (0.1 M), and centrifuged for 10 min. The supernatant (coming from GH3 or mouse pituitary lysate) was run directly or stored frozen (−80 °C) for later analysis. The total cAMP was determined with the ELISA kit following the manufacturer's instructions.

**Quantification of inositol monophosphate.** Changes in second messenger inositol monophosphate were quantified on cell lysates using the IP-One Enzyme-linked Immunosorbent Assay (ELISA) assay kit (Cisbio, Codolet, France, Cat. No 72IP1PEA/D) according to the manufacturer's instructions. WT or CRISPR generated HEK ($\Delta G_{s/olf}$, $\Delta G_{q/11}$, $\Delta G_{12/13}$, and $\Delta G_{tot}$) or GH3 cells were seeded into a 24-well plate (100,000 cells per well) and after overnight incubation, they were transfected with increasing concentrations (50, 150, and 300 ng) of empty vector (MOCK) or pcDNA3.1 FLAG-GPR101, by using X-tremeGene 9 DNA transfection reagent (Roche, Cat. No 39320900). Forty-eight hour post-transfection, cells were stimulated with vehicle or GHRH (10 nM) when needed, and culture medium was replaced by 200 µl of provided stimulation buffer and returned to the incubator for 1 h. Cells were then lysed by adding 50 µl of lysis reagent (2.5%) and further incubated for 30 min at 37 °C. Finally, 50 µl of cell lysate was transferred into the ELISA plate, and the assay was conducted according to the manufacturer's instructions. Pituitary tissues from 29-week-old $Ghrhr^{Gpr101}$ and WT mice were also collected and lysed as the cells and were assayed following the manufacturer's instructions.

**Measurement of TGFα shedding.** WT or CRISPR generated HEK $\Delta G_{q/11}$ or $\Delta G_{12/13}$ cells were transfected using X-tremeGene 9 DNA transfection reagent (Roche, Cat. No 39320900) with 36 ng of pCAGGS/AP-TGFα and 14 ng of receptor (pcDNA3.1 FLAG-GPR101 or pcDNA3.1 FLAG-β2AR) diluted 1/5, 1/2 or not diluted with pcDNA3.1 empty vector to obtain different amounts of receptor expression. Twenty-four hour later, 80 µl of conditioned medium was transferred into a new transparent flat-bottom 96-well plate. When both plates had cooled down to room temperature, 80 µl of freshly prepared AP solution (1.2 ml Tris-HCl 2 M pH 9.5, 0.2 ml NaCl 4 M, 0.2 ml MgCl₂ 1 M, 18.4 ml H₂O, 200 µl pNPP 1 M) was added into each well. Optical density (OD) at 405 nm was measured directly and after 1 h of incubation using the WALLAC VICTOR 2 microplate reader (Perkin Elmer Life Sciences). We calculated relative the percentage of AP activity in conditioned medium: $AP = \Delta OD_{405}\,CM/(\Delta OD_{405}\,CM + \Delta OD_{405}\,Cell)$, where $\Delta OD_{405}\,CM$ and $\Delta OD_{405}\,Cell$ denote changes in $OD_{405}$ in the conditioned medium (CM) and on the cell surface, respectively, before and after a 1 h incubation in the presence of pNPP. The relative percentage of AP activity in conditioned medium was normalized by the relative percentage of AP activity obtained for well transfected with pCAGGS/AP-TGFα and pcDNA3.1 empty vector instead of receptor. The same experiment was performed in CRISPR generated HEK $\Delta G_{tot}$ cells where GPR101 was co-transfected (by using the X-tremeGene 9 DNA transfection reagent (Roche, Cat. No 39320900) with empty vector or each of the G proteins ($G_{\alpha s}$, $G_{\alpha q}$, $G_{\alpha 11}$, $G_{\alpha 12}$, or $G_{\alpha 13}$) or chimeric G proteins ($G_{\alpha qs}$, $G_{\alpha qi1}$, $G_{\alpha qi3}$, $G_{\alpha qo}$, and $G_{\alpha qz}$).

**Rho activation assay.** WT or $\Delta G_{tot}$ HEK293 cells were seeded in 10-cm culture dishes and transfected with 15 ug of pcDNA3.1 FLAG-GPR101 (or empty vector, MOCK). After 24 h, cells were starved overnight with FBS-free medium. Cells were rinsed with ice-cold TBS (25 mM Tris-HCl pH 7.5, 150 mM NaCl), scraped with cold lysis/binding/wash buffer (0.5 ml) containing protease inhibitors (Roche, Basel, Switzerland, cOmplete Tablets Mini EDTA-free, EASYpack, Cat. No 04693159001) and incubated on ice for 5 min. Supernatants were obtained by centrifugation at 16,000 × g for 15 min at 4 °C, and total protein concentration was determined by the bicinchoninic acid (BCA) assay (Thermo scientific, USA, Pierce™ Protein Assay Kit, Cat. No 23227). Cell lysates (500 µg) were immuno-precipitated with a GST-tagged Rhotekin Rho-binding domain (RBD) bound to agarose beads and the abundance of active GTP-bound Rho (A, B, and C) was analyzed by pull-down assay, by using the Rho activation assay kit (Enzo, Cat No ADI-EKS-465).

**Determination of ERK phosphorylation.** WT or CRISPR generated HEK $\Delta G_{s/olf}$, $\Delta G_{q/11}$, $\Delta G_{12/13}$, and $\Delta G_{tot}$ cells were seeded in 35-cm culture dishes and transiently transfected with empty vector (MOCK) or pcDNA3.1 FLAG-GPR101. Twenty-four hour later, cells were starved overnight with medium containing 1% FBS. Cells

were immediately put on ice and lysed with cold RIPA Buffer (25 mM Tris–HCl, 150 mM NaCl, 1% NP-40, 1% sodium deoxycholate, 0.1% SDS; pH 7.6) containing protease inhibitors (Roche, Basel, Switzerland, cOmplete Tablets Mini EDTA-free, EASYpack, Cat. No 04693159001) and phosphatase inhibitors (Roche, Basel, Switzerland, PhosSTOP, EASYpack, Cat. No 04906837001) for 20 min. Lysates were centrifuged at 15,000 × g for 15 min at 4 °C. Cell lysates (with an equal amount of protein (20 µg) as detected by bicinchoninic protein assay (BCA, Thermo scientific, USA, Pierce™ Protein Assay Kit, Cat. No 23227) were mixed with SDS-PAGE reducing sample buffer, boiled at 95 °C, and separated by 10% SDS-PAGE followed by immunoblotting (transfer to nitrocellulose membrane (Amersham Protran 0.2 NC 300 mm × 4 m, Cat. No 10600001). The membrane was blocked in Tris-buffered saline containing 0.1% Tween (TBS-T) and 5% fat-free dry milk powder (Sigma–Aldrich, Skim Milk powder, Cat. No 70166) for 60 min at room temperature. Then, the membrane was incubated in TBS-T containing 5% milk and the primary antibodies overnight at 4 °C. ERK$_{1/2}$ phosphorylation was detected with an anti-phospho-p44/42 MAPK (ERK$_{1/2}$) (Y204/Y204, D13.14.4E) rabbit monoclonal antibody (Cell Signaling Technology, Cat. No 4370 S, 1:2000), anti-p44/42 MAPK (ERK$_{1/2}$) rabbit monoclonal antibody (Cell Signaling Technology, Cat. No 4695 S, 1:1000). After washing with TBS-T (3 times), the membrane was incubated in TBS-T containing 5% milk and a secondary anti-rabbit IgG horseradish peroxidase (HRP)-linked antibody (Cell Signaling Technology, Cat. No 7074 S, 1:1000 dilution) for 60 min at room temperature. The membrane was washed in TBS-T, developed for 2 min in a mix (50:50) of Detection Reagent 1 Peroxide Solution (Thermo scientific, USA, cat. No 1859700) and Detection Reagent 2 Luminol Enhancer Solution (Thermo scientific, USA, cat. No 1859697). Immunoreactive signal was quantified and signal levels of Phospho-ERK were normalized to total ERK1/2.

**Co-immunoprecipitation of GPR101 receptor with the G proteins.** HEK293 cells were seeded in 10-cm culture dishes and co-transfected with 15 µg of FLAG-GPR101 (or empty vector, MOCK) and 15 µg of HA-tagged G-protein ($G_{\alpha s}$, $G_{\alpha q}$, $G_{\alpha 11}$, $G_{\alpha 12}$, $G_{\alpha 13}$, $G_{\alpha i1}$ or empty vector, MOCK) expression vectors. 24 h later, co-transfected HEK cells were serum starved overnight. Medium was removed and cells were washed twice in ice-cold PBS and homogenized in 500 µl of cold lysis buffer (20 mM Tris-HCl pH 7.5, 100 mM NaCl, 10 mM MgCl₂, 2 mM EDTA, 1% Triton-X100, 1% protease and phosphatase inhibitor cocktail). The homogenates were placed on a rotator for 30 min at 4 °C and clarified by centrifugation for 10 min at 10,000 × g and 4 °C. An aliquot of the supernatant was reserved for analysis of the total cell lysate (Input) and the remaining sample (900 µg) was used for co-immunoprecipitation with 5 µl of a mouse anti-FLAG antibody (Cell Signaling Technology, Cat. No 8146) for 1 h at 4 °C. After conjugation, Protein A/G PLUS-Agarose beads (Santa Cruz, Cat. No 2003) were washed and added to the samples for 1 additional hour on a rotator at 4 °C. Immune complexes were subsequently washed three times with PBS containing 1% NP-40 and protease and phosphatase inhibitors, and were resuspended in 100 µl of 1 × SDS buffer and incubated for 1 h at room temperature. After centrifugation, the supernatant was removed and directly loaded onto a Tris glycine 4–10% SDS-PAGE gel (100 V, 2 h). Proteins were then transferred to nitrocellulose membranes (100 V, 90 min) and probed overnight at 4 °C with a rabbit anti-HA antibody (Cell Signaling Technology, Cat. No 3724, dilution 1:1000) against the HA-tagged $G_\alpha$ proteins. The primary antibody was detected using a secondary anti-rabbit IgG horseradish peroxidase (HRP)-linked antibody (Cell Signaling Technology, Cat. No 7074 S, 1:1000 dilution) for 1 h and the membranes were developed and images were acquired using the ImageQuant LAS4000 system.

**Determination of PKA and PKC phosphorylation.** GH3 cells were seeded in 35-cm culture dishes and transiently transfected with empty vector (MOCK) or pcDNA3.1.FLAG-GPR101. 24 h later, cells were starved overnight with medium containing 1% FBS, and then harvested by using ice-cold lysis buffer (20 mM HEPES, 1% NP-40, 0.1% SDS, 150 mM NaCl, 2 mM EDTA) containing protease and phosphatase inhibitors for 20 min. Lysates were centrifuged at 15,000 × g for 15 min at 4 °C. The protein concentration was determined by the BCA assay. Cell lysates (30 µg) were subjected to western blot analysis using conventional SDS-PAGE (10%) and protein transfer onto nitrocellulose filters. The membrane was blocked with TBS-T containing 5% Bovine Serum Albumin (BSA, Sigma–Aldrich, Cat. No A9647) for 60 min at room temperature. Then, the membrane was incubated in TBS-T containing 5% milk and the primary antibodies overnight at 4 °C. Primary antibodies were used as described in the manufacturer's protocols: anti-phospho-PKC-α/βII (Thr638/641) rabbit monoclonal antibody (Cell Signaling Technology, Cat. No 9375 S, dilution 1:1000); anti-PKC-α rabbit monoclonal antibody (Cell Signaling Technology, Cat. No 2056 S, dilution 1:1000); anti-phospho-PKA(Thr197) mouse monoclonal antibody (Cell Signaling Technology, Cat. No 4781 S, dilution 1:1000); anti-PKA-α rabbit polyclonal antibody (Cell Signaling Technology, Cat. No 4782 S, dilution 1:1000). After washing with TBS-T (3 times), the membrane was incubated in TBS-T containing 5% BSA and peroxidase-conjugated secondary antibody (anti-rabbit IgG horseradish peroxidase (HRP)-linked antibody, Cell Signaling Technology, Cat. No 7074 S, dilution 1:1000; anti-mouse IgG horseradish peroxidase (HRP)-linked antibody, Cell Signaling Technology, Cat. No 7076P2, dilution 1:1000) for 60 min at room temperature. After washing, proteins were visualized with detection reagents

(Detection Reagent 1 Peroxide Solution and Detection Reagent 2 Luminol Enhancer Solution) and western blot signals were quantified with ImageJ v.1.47 (wayne Rasband, National Institute of Health, USA). Signal levels of phospho-PKC and phospho-PKA were normalized to total PKC and PKA, respectively. The same experiment was repeated with incubation of GH3 cells and the siRNAs targeting $G_{\alpha s}$, $G_{\alpha q/11}$, $G_{\alpha 12/13}$ or the nontargeting (NTS) (1 μM) (according to the manufacturer's instructions) in the siRNA delivery medium. After 24 h, cells were transiently transfected with empty vector (MOCK) or pcDNA3.1 FLAG-GPR101 (using lipofectamine 3000). Forty-eight hour later, cells were harvested and proteins were detected by western blot.

**Growth hormone secretion**. For in vitro GH secretion, GH3 cells were seeded on 24-well plates and transfected with MOCK or GPR101 or GHSR or GHRHR expression vector. After 24 h, the cells were starved for 3 h at 37 °C and GHSR and GHRHR were stimulated with vehicle or their agonists GHRH (10 nM) and GHS (10 nM), respectively (for 15 min at 37 °C). After incubation, the medium was collected at different time points (0, 1, 2, and 6 h) and centrifuged at 12,000 × g for 10 min. The concentrations of GH in the supernatant were determined using the Growth Hormone Rat ELISA Kit (Merck Millipore, Cat. No EZRMGH-45K) according to the manufacturer's instructions. The same experiment was repeated with incubation of GH3 cells and the siRNAs targeting $G_{\alpha s}$, $G_{\alpha q/11}$, $G_{\alpha 12/13}$ or the nontargeting siRNA (NTS) (according to the manufacturer's instructions) in the siRNA delivery medium. After 24 h, cells were transiently transfected with empty vector (MOCK) or pcDNA3.1.FLAG-GPR101 or pcDNA3.1.FLAG-GHRHR or pcDNA3.1.GHSR (using lipofectamine 3000). 48 h later, cells were starved for 3 h at 37 °C and GHSR and GHRHR were stimulated with vehicle or their agonists GHRH (10 nM) and GHS (10 nM), respectively (for 15 min at 37 °C). After incubation, the media were collected at different time points (0, 1, 2, and 6 h) and centrifuged at 12,000 × g for 10 min. The concentrations of GH in the supernatant were determined by using the GH Rat ELISA Kit according to the manufacturer's instructions. In case of using molecules, GH3 cells were seeded in 24-well plates and transfected with empty vector (MOCK) or pcDNA3.1 FLAG-GPR101. After 24 h, cells were treated with Calphostin C (10 μM) or H89 (10 μM). The day after, cells were then starved at 37 °C and at the end of incubation, the medium was removed at different time points (0, 1, 2, and 6 h) and centrifuged at 12,000 × g for 10 min, and the supernatants were used for the hormone assay. Secreted GH in the medium was measured using the Growth Hormone Rat ELISA Kit.

**XTT cell proliferation assay to measure cell growth**. GH3 cells were seeded in 96-well flat-bottom microplates. Twenty-four hour later, cells were transfected with increasing concentrations (0, 25, 50, and 100 ng) of empty vector or GPR101 or GHRHR or GPR101 + GHRHR. Then, the cells were stimulated with various amounts of vehicle or GHRH (0–300 nM) (final volume of 100 μl per well) at 37 °C for 24 h. After the incubation period, 50 μl of the XTT (sodium 3′-[1-[(phenyla-mino)-carbony]-3,4-tetrazolium]-bis(4-methoxy-6-nitro)benzene-sulfonic acid hydrate) labeling mixture (mix 5 ml XTT labeling reagent with 0.1 ml electron coupling reagent) were added to each well, and the microplate was incubated for 4 h at 37 °C and 5% $CO_2$. After this incubation period, the formazan dye formed is quantified by measuring the optical density of the samples using the WALLAC VICTOR 2 microplate reader (Perkin Elmer Life Sciences). The wavelength to measure absorbance of the formazan product is 450 nm. For the use of siRNAs, GH3 cells were plated in 96-well plates and allowed to adhere overnight. Accell siRNAs ($G_{\alpha s}$, $G_{\alpha q/11}$, and $G_{\alpha 12/13}$) were added to Accell siRNA delivery medium (supplemented with 2% FBS) for a final concentration of 1 μM. 100 μL of the Accell siRNA and medium mixture was then added (per well) to the cells after the growth medium had been aspirated. Cells were incubated for 24 h at 37 °C and 5% $CO_2$, and then transfected with MOCK, GPR101, GHRHR or GHSR for an additional 24 h. After this incubation time, cells were respectively treated with vehicle or GHRH (10 nM) or GHS (10 nM) for an additional 24 h, and GH3 viability was assessed by the XTT method. To study the effects of different molecules, rat GH3 cells were cultured and transfected with MOCK or GPR101. After 48 h, cells were treated with vehicle (0.1% DMSO), H89 (10 μM), 8-Br-cAMP (10 μM), FSK (10 μM), or calphostin C (10 μM) overnight at 37 °C. To terminate experiments, XTT was directly added to the culture media in order to measure cell growth.

**Animals**. All mice were bred and maintained on a C57BL/6 J genetic background and were housed in standard cages under specific pathogen-free conditions, fed standard mouse chow and water ad libitum and kept on a 12 h light/dark cycle. All experiments were approved by the animal care and use committee of the University of Liège under the accredited protocol number 1812 and 1776.

**Construction of the promGHRHR-FLAG-Gpr101 transgene and generation of transgenic $Ghrhr^{Gpr101}$ mice**. A 1948-bp rat GHRHR promoter sequence[64] (NCBI, accession number AF 121969.1) was obtained from rat tail genomic DNA (purification by the Quick DNA Universal Kit, Zymo Research, Cat. No D4068) and amplified by Polymerase Chain Reaction (PCR) using the following primers: Forward primer, 5′-GAGAGGATCCCCATGGCCTCTGCATCAACTTCTG-3′; Reverse primer, 5′-GAGACTCGAGCTGTAGTCCGCCCCAAAGAG-3′. Then, the PCR-amplified 1948-bp GHRHR promoter was subcloned into the pcDNA3.1

plasmid by using the BamHI and XhoI restriction sites, and its authenticity was verified by DNA sequencing. The 1587pb- mouse Gpr101 construct following a FLAG epitope and preceding a poly-adenylation (poly A) signal sequence was amplified by PCR from the pcDNA3.1 FLAG-Gpr101 and flanked by XhoI and XbaI cloning sites with the primers: Forward primer, 5′-GAGACTCGA-GaccATGGATTATAAAGATGATGATAAA-3′; Reverse primer, 5′-GAGAccTCTAGAccTTAAGGTGAAGTAGCTGAATCATG-3′. Subsequently, the XhoI-XbaI fragment (containing FLAG-Gpr101-Poly A) was transferred into the pcDNA3.1 plasmid downstream of the GHRHR promoter sequence. Its authenticity was also verified by DNA sequence analysis. The linear 3871bp- transgene DNA (promGHRHR-FLAG-Gpr101-poly A) was obtained by PCR and purified by using Wizard® SV Gel and PCR Cleanup System (Promega, Cat. No A9281) and Minielute® Reaction Cleanup Kit (Qiagen, Cat. No28204). This highly purified DNA (20 ng μl$^{-1}$) was microinjected into the pronuclei of fertilized eggs of C57BL/6 J mice at the GIGA-Transgenesis platform of the university of Liège (Belgium). Microinjected eggs were transferred into the uteri of female mice and allowed to develop to term.

**Immunofluorescence of pituitary tissues**. Pituitaries from 29-week-old $Ghrhr^{Gpr101}$ and WT mice were fixed overnight with 4% paraformaldehyde (PFA) at 4 °C and paraffin-embedded. Then, tissues were sectioned at 5 μm and mounted on Superfrost® Microscope Slides (Thermo Scientific™, Cat. No 12372098). After deparaffinization and antigen heat retrieval (using citrate buffer at pH 6), sections were washed with PBS, and permeabilized at room temperature for 10 min with PBS containing 0.5% Triton X-100. After wash, they were blocked in blocking buffer (PBS containing 5% FBS and 0.5% Triton X-100) for 60 min, and then incubated with a primary antibody [polyclonal rabbit anti-GHRHR (LSBio, Cat. No LS-B6566, dilution 1:100) or rabbit anti-Pit-1 (Novus Biologicals, Cat. No NBP1-92273, dilution 1:500) or goat anti-GH (R&D systems, Cat. No AF1067-SP, dilution 1:133) antibody] overnight at 4 °C. Sections were subsequently washed three times in PBS containing 0.5% Triton X-100, prior to a 2 h room temperature incubation with the secondary antibody [anti-rabbit IgG (H + L) F(ab′)$_2$ fragment Alexa Fluor 647 conjugate (Cell Signaling Technology, Cat. No 4414, dilution 1:1000) or anti-goat Alexa Fluor 488 conjugate (Abcam, Cat. No ab150129, dilution 1:200) antibody]. After, sections were washed three times in PBS containing 0.5% Triton X-100 and incubated with a monoclonal mouse anti-FLAG antibody (Sigma–Aldrich, Cat. No F3165, clone M2, dilution 1:1000) for 2 h at room temperature. Sections were again washed three times in PBS containing 0.1% Triton X-100 and incubated with the secondary antibody [anti-mouse IgG Fab2 Alexa Fluor 488 conjugate (Cell Signaling Technology, Cat. No 4408 S, dilution 1:1000) or anti-mouse-IgG-Atto 647 N conjugate (Sigma–Aldrich, Cat. No 50185, dilution 1:200) antibody] for 2 h at room temperature. Sections were then washed three times in PBS containing 0.5% Triton X-100 and mounted using ProLong Gold Antifade Mountant containing DAPI (ThermoFisher Scientific, Cat. No P36931). Finally, stained pituitaries were visualized by confocal microscopy and image acquisition was performed on a NIKON A1R (Tokyo, Japan) confocal microscope (oil immersion objective, ×60 magnification).

**Determination of plasma hormones**. The blood of 6-, 26-, and 52-week-old $Ghrhr^{Gpr101}$ and WT mice were collected from the inferior vena cava in EDTA capillary blood tubes (Greiner, Cat. No 450475) and the plasma was separated from cells by centrifugation (2000 × g) for 10 min. Plasma hormone levels were determined using commercial ELISA immunoassays according to the protocol of the manufacturers. Mouse GH was detected and quantified by using the Rat/Mouse Growth Hormone ELISA Kit (Merck Millipore, Cat. No EZRMGH-45K). Mouse PRL was detected and quantified by using the PRL Mouse ELISA Kit (Thermo-Fisher Scientific, Cat. No EMPRL). Mouse IGF-1 was detected and quantified by using a Mouse/Rat IGF-I Quantikine ELISA Kit (Biotechne, Cat. No MG100).

**Pituitary superfusion assay**. Pituitary glands from 29-week-old $Ghrhr^{Gpr101}$ and WT mice were excised and immediately washed in superfusion chambers (0.5 ml volume) for 30 min at 37 °C in oxygenated Dulbecco's Modified Eagle's Medium (containing L-glutamine, 4.5 g/L glucose, 25 mM HEPES) supplemented with BSA (0.1%). Then, they were superfused with the same medium (at a rate of 0.1 ml min$^{-1}$). After 2 h of equilibration, effluents (500 μl) were collected every 5 min. Mouse GHRH (100 nM) and KCl (0.03 M) were added to the medium for 15 and 20 min, respectively, separated by superfusion with medium alone. Samples were stored at −80 °C until GH determination by the Rat/Mouse Growth Hormone ELISA Kit (Merck Millipore, Cat. No EZRMGH-45K). For the quantification of total protein concentration, WT and $Ghrhr^{Gpr101}$ pituitary glands were excised weighted (by an analytical balance) and lysed in RIPA buffer (Invitrogen, Cat. No 89900) containing protease inhibitors for 30 min at 4 °C. Protein extracts were then obtained by centrifugation (25,200 × g, 15 min, 4 °C) to remove tissue debris. The BCA protein assay (Thermo scientific, USA, Pierce™ Protein Assay Kit, Cat. No 23227) was used to measure total protein concentrations (in pituitaries) compared to a protein (BSA) standard.

**Histology of mouse pituitaries and fat**. Mouse pituitaries and epididimal white fat were isolated from 27-week-old $Ghrhr^{Gpr101}$ and WT mice, washed in PBS and

fixed in 4% PFA at 4 °C overnight. After that they were embedded into paraffin, cut with a microtome (5 μm section), mounted on Superfrost® Microscope Slides and stained with H&E (Merck, Cat. No 1.05174.1000 and 1.09844.1000, respectively). Photos were taken with a FSX100 microscope equipped with a camera (Olympus, USA) using ImageJ v.1.47 (wayne Rasband, National Institute of Health, USA). Adipocyte size (or area) was calculated using Adiposoft v1.15 plug-in software (Center for Applied Medical Research CIMA, University of Navarra, Spain).

**Reticulin staining**. Five micrometer paraffin-embedded pituitary sections of 27-week-old $Ghrhr^{Gpr101}$ and WT mice were stained with the Reticulum Stain Kit (Sigma–Aldrich, Cat. No HT102A-1KT) as recommended by the manufacturer.

**Oil Red O staining**. Liver tissues were isolated from the 27-week-old $Ghrhr^{Gpr101}$ and WT mice, washed in PBS and fixed in 4% PFA at 4 °C overnight. After that, tissues were immersed in 30% sucrose overnight and they were cut with a cryostat (7μm thickness), mounted on Superfrost® Plus Microscope Slides (Thermo Scientific™, Cat. No J1800AMNT), and stained with Oil Red O (ORO) (Sigma–Aldrich, Cat. No O0625) following the manufacturer's protocols. Photos were taken with the FSX100 microscope.

**Detection of the Ki-67 cell proliferation marker by immunohistochemistry and immunofluorescence**. After paraffin removal and antigen retrieval (using citrate buffer at pH 6), pituitary sections from 27-week-old $Ghrhr^{Gpr101}$ and WT mice were incubated in Hydrogen Peroxide (H$_2$O$_2$, 0.3%) for 10 min to block endogenous peroxidase activity. After that, sections were stained with rabbit polyclonal anti-Ki-67 antibody (Merck, Cat. No AB9260) (dilution 1:300) by using the Rabbit Specific HRP/DAB (ABC) Detection IHC kit (abcam, Cat. No ab64264) according to the manufacturer's protocol. Photos were taken with the FSX100 microscope. The Ki-67 index was quantified by determining the number of Ki-67 positive cells among the total number of cells. In the immunofluorescence protocol, pituitary sections were blocked in 5% FBS and incubated with a goat anti-GH polyclonal antibody overnight at 4 °C. Then, sections were washed and incubated with an anti-goat AF 488 conjugate antibody (dilution 1:200) for 2 h at room temperature. After washing, sections were incubated with the rabbit anti-Ki-67 antibody (dilution 1:300) for 2 h at room temperature. Sections were again washed and incubated with an anti-rabbit AF 647 conjugate antibody (dilution 1:1000) for 2 h at room temperature. Finally, sections were washed in PBS, mounted on slides, air-dried overnight and visualized by using the NIKON A1R confocal microscope (×60 magnification).

**Length and weight of body, organs and bones**. Body weight of female and male $Ghrhr^{Gpr101}$ and WT mice was measured once a week (starting at 3 weeks and ending at 53 weeks) by using a weighing scale. Tail length as well as mouse body length (from the tip of nose to the anus) were measured with a caliper under anesthesia (4% isoflurane) from 3 weeks until 69 weeks of age. For the femur, 27-week-old $Ghrhr^{Gpr101}$ J mic/J mice were euthanized and left and right femurs were removed and dissected free of soft tissue. Femur lengths were measured using a micrometer caliper. The organ (liver, kidney, heart, lung, testis, and epididymal fat) weights were measured with an analytical balance.

**MicroCT imaging**. The 27-week-old $Ghrhr^{Gpr101}$ and WT mice underwent an in vivo X-ray computed tomography images to assess the mice body composition in terms of volume. The CT scans were acquired on an eXplore 120 micro-CT (Gamma Medica, USA/GE Healthcare, UK) with a customized protocol (70 kV, 0.512 mAs, 360 views over 360°, continuous rotation) provided by the manufacturer. The software for data collection was Host Console Interface and Micro-View ABA 2.3.a7. During the imaging session, mice were under general anesthesia by isoflurane (in a mixture of 30% of O$_2$ in air), placed in prone position in a dedicated animal holder equipped with an air warming system (Equipment Minerve, Esternay, France). The mice were continuously monitored (respiratory rate and temperature). All micro-CT images were reconstructed using the Feld Kamp's filtered back projection algorithm with a cutoff at the Nyquist frequency to obtain a 3D volume with an isotropic voxel size of 100 μm. In order to assess the CT signal intensity of the adipose tissue, CT scans of different freshly harvested fat types (epididymal white fat, subcutaneous white fat, adrenal white fat, and brown fat) were also acquired.

**Assessment of body composition**. CT images were used to assess the fat volume as well as lean volume. For this aim a semi-automated segmentation procedure was employed using PMOD 3.6 software (PMOD Technologies, Zurich, Switzerland; RRID:SCR_016547). Briefly, an intensity threshold range (−280 to −160 HU (Hounsfield unit)) for fat was obtained based on the images of the ex vivo harvested fat types. In vivo CT image of the whole mouse was first manually segmented to remove the bed and all extra signal (e.g. tubes of warming system). The resulting image was then sent to the PMOD automated segmentation to extract the fat part. Based on the mean signal intensity (threshold for bone: 250 HU), extracted on a spherical region of interest placed on the bone, binary mask of the bone was extracted using automated

segmentation method implemented in PMOD. A total body mask was also generated using a range of signal intensities containing fat, bone, and muscle. Knowing the voxel size and the number of voxels in the obtained masks, the volume of fat, bone, and total body were calculated. The % of Fat mass was calculated as follow:

$$\% \text{ Fat mass} = \text{Fat volume} \times 100/\text{Total body volume}$$

$$\% \text{ Bones} = \text{Bone volume} \times 100/\text{Total body volume}$$

$$\% \text{ Lean mass} = [\text{Total body volume} - (\text{Bone volume} + \text{Fat volume})] \times 100/\text{Total body volume}$$

We also used the segmented bone images to measure the length of bones such as tibia, femur, humerus, and ulna, as well as skull and pelvic dimensions.

**Glucose tolerance test**. Female and male $Ghrhr^{Gpr101}$ and WT mice (11 months) were used for the glucose tolerance test after a 12h-lasting overnight food-withdrawal (overnight fasting). Blood glucose levels were measured by collecting one drop of blood from the tail and using the Accu-Chek® Aviva glucose analyzer (Roche, Mannheim, Germany, Cat. No 06988563016). For the glucose tolerance test, mice were administered IP with D-glucose (2 g kg$^{-1}$ body weight), and the blood glucose levels were measured at 30, 60, and 90 min after glucose injection. Upon completion of the experiment, mice were placed in a cage supplied with food and water.

**Activation of PKC in human GH-secreting pituitary adenomas**. Human GH-secreting (GH-omas) pituitary adenomas were obtained by surgery from acromegalic patients (with and without *AIP* mutations), and from patients with X-LAG. Patients provided written informed consent and the study was performed with the approval of the Ethical Committee of the Centre Hospitalier Universitaire de Liège. Pituitary tumors were fixed with 4% PFA at 4 °C overnight and paraffin-embedded. Tissues were sectioned at 5 μm and mounted on Superfrost glass slides. Paraffin-embedded sections from mouse pituitaries (29-week-old $Ghrhr^{Gpr101}$ J mic/J mice), human aryl hydrocarbon receptor interacting protein (AIP) mutated GH-omas, human X-LAG tumors and human GH-omas without an underlying genetic cause were deparaffinized and antigen retrieval was performed using citrate buffer (pH 6). Sections were washed with PBS, and permeabilized at room temperature for 10 min with PBS containing 0.5% Triton X-100. After wash, sections were blocked in blocking buffer (PBS containing 5% FBS and 0.5% Triton X-100) for 60 min. For PKC activation, sections were incubated with an anti-phospho-PKCα (Thr638) rabbit polyclonal antibody (Life Technologies, Cat. No 44-962 G) diluted (1:500) in blocking buffer overnight at 4 °C. Sections were subsequently washed three times in PBS containing 0.5% Triton X-100, prior to the incubation with the secondary antibody, anti-rabbit IgG Fab2 Alexa Fluor 647 (#4414 S, Cell Signaling Technology, Danvers, MA, USA) diluted (1:1000) in blocking buffer for 2 h at room temperature. Sections were washed three times in PBS containing 0.5% Triton X-100 and mounted using ProLong Gold Antifade Mountant containing DAPI (Molecular Probes, ThermoFisher Scientific, Waltham, MA, USA, Cat. No P36931). Cells were visualized by confocal microscopy and image acquisition was performed on a confocal microscope NIKON A1R (oil immersion objective ×60, Tokyo, Japan).

**Data analysis**. Data were analyzed using GraphPad Prism v.6 (GraphPad Software, San Diego, CA, USA), Microsoft Excel (Microsoft Office, Microsoft®, USA, version 16.16.24 (200713)) and ImageJ v.1.47 (National Institutes of Health, USA) bundled with Java 1.8.0_172. Statistical analysis was performed with statistical significance determined as follow: not significant (ns) $p > 0.05$, *$p < 0.05$, **$p < 0.01$, and ***$p < 0.001$. If the data followed a Gaussian distribution, we compared them using unpaired $t$-test. However, if the normal distribution of data or the homogeneity of their standard deviation was not verified, we compared them using the nonparametric Mann–Whitney test.

**Reporting summary**. Further information on research design is available in the Nature Research Reporting Summary linked to this article.

## Data availability
All the data and materials are available from the authors upon reasonable request. Source data are provided with this paper.

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

## Acknowledgements

This work was supported by the Fonds pour la Recherche Scientifique (F.R.S.-FNRS) Incentive Grant for Scientific Research (F.4510.14) and Research Project (PDR T.0111.19), Télévie (7461117 F, 7454719 F), University of Liège (Action de Recherche Concertée ARC 17/21-01), and Léon Fredericq Foundation. D.A. was supported by a postdoctoral In WBI fellowship and is a Télévie fellow. J.H. and A.P. are F.R.S.- FNRS Research Associate and Research Director, respectively. N.D. was supported by a FRIA PhD fellowship. A.I. was funded by the PRIME (JP18gm5910013) and the LEAP (JP18gm0010004) from the Japan Agency for Medical Research and Development (AMED); JSPS KAKENHI grant (17K08264) from Japan Society for the Promotion of Science. A.F.D. and A.B. were supported by grants (to AB) from the Fonds d'Investissement Pour la Recherche Scientifique (FIRS) of the Centre Hospitalier Universitaire de Liège and from the JABBS Foundation, UK. A.C. was supported by the Luxembourg Institute of Health (LIH) and Luxembourg National Research Fund (PRIDE-11012546 "NextImmune). J.H. and A.C. are members of the "European Research Network on Signal Transduction" (ERNEST, COST action CA18133). We thank the GIGA imaging platform for the technical support in confocal image acquisition and FACS analysis, the Histology platform for tissue preparation, the animal facility, the GIGA Transgenics platform as well as the GIGA-CRC in vivo Imaging platform.

## Author contributions

Initiated the project and attracted funding: J.H. and A.B., Formulated study hypotheses: D.A., A.F.D., J.H., and A.B., Experimental study design and supervision: J.H., Performed experiments: D.A, Contributed human pituitary samples: A.F.D. and A.B., Analyzed the data and constructed the figures: D.A. and J.H., Generated new research reagents: A.I. and A.C., Design and generation of the transgenic mouse *Ghrhr^Gpr101*: J.H., D.A., and F.E., C.T. scan data acquisition and analysis: D.A., M.B., and A.P., GPR101 cloning and HEK293 IP$_1$ experiments: N.D., Wrote all versions of the manuscript: D.A., A.F.D., J.H., B.P., and A.B.

## Competing interests

The authors declare the following competing interests A.F.D. and A.B. are Inventors of granted US Patent No. 10,350, 273B2, Treatment of Hormonal Disorders of Growth. All other authors declare no competing interests.
