## [Peer Review File · Nature Communications]

Reviewers' comments, first round:

Reviewer #1 (Remarks to the Author):

Abboud et al describe a new mouse model (tg-GHRH-GPR101) and interesting new findings on how GPR101 may mediate its signaling. Unfortunately, the mouse data do not replicate the human disease (which is due to pituitary hyperplasia or tumor/adenoma formation) and the logic behind using the GHRH promoter to guide GPR101 expression to replicate the human condition is not very clear. As a model, nevertheless, the data are interesting and the signaling findings are also of interest to the field.

Here are some additional specific comments:

1. There is no obvious strong GPR101 co-localization with GH (Fig. 1A) or Pit-1 (Suppl Fig. 1D). In the latter, GPR101 expression does not seem to be in the membrane as in the other figures (Fig. 1A and Suppl Fig. 1C). Better images for GPR101-Pit-1 have to be obtained. Moreover, zoom-in boxes at higher magnification would be useful in all the figures. The lack of clear GPR101-GH co-localization mimics what seen in patients with X-linked acrogigantism (X-LAG). It is thus important to confidently show whether GPR101 is expressed in progenitors/undifferentiated cells, likely of the Pit-1 lineage, by showing better GPR101-Pit-1 co-localization as well as co-localization (or lack of) with other markers (e.g. Sox2, S100, etc). GPR101 co-localization with GHRHR needs also to be tested to prove the cell type specificity of the rat *Ghrhr* promoter sequence used.
2. GPR101 over-expression attained in the pituitary of TG mice (about 20 fold increase compared to controls) is lower than what seen in the tumors of X-LAG patients (up to hundreds-fold increase). This difference in magnitude could also be a factor explaining, maybe in part, the lack of pituitary tumorigenesis observed in the TG mice.
3. The results of mouse experiments (Fig. 1-3, and especially for Fig.1) should be broken down by sex. The authors do not specify if they analyzed both sexes or just males or females. If the numbers are small, experiments need to be expanded.
4. The authors somewhat ignore the fact that their mouse model got much more significant hyperPRLmia (Figure 1 D) than either GH or IGF1. In the text (page 8), they say "in addition to hyperPRLmia", but in fact hyperPRLmia was the main abnormality in addition to hyperGH...This finding is not adequately explained.
5. Was high PRL responsive to dopamine agonists?
6. In addition to body length, if available, the authors should also show whether there were any differences in tail length.
7. Growth curves for height and weight should be used in Fig. 1F-G in place of just one time point. In particular, mouse height shown in Fig. 1F, is not very convincing. 2 cluster of data can be seen. Do they represent males vs females or mice from different litters?
8. RNA-seq data from mouse TG pituitaries would be very useful to confirm at a transcriptome pathway level the effects seen on signalling/proliferation/GH transcription (Fig. 4-6). Moreover, they can aid in the discovery of possible new pathways causing the observed phenotype.
9. The results showing new signalling pathways activated by GPR101 are very interesting. Did the authors try to test the effect of the putative GPR101 ligand, GnRH(1-5)?
10. Does GPR101 co-immunoprecipitate with the various G-proteins mentioned?

11. Could mathematical modelling (in silico) predict these non-Gs interactions?

Other minor comments:

1. Page 8: "...tibial length was markedly increased": delete "markedly"

2. Abstract and elsewhere: "prosecretory role" what is the meaning of this term? 'prosecretory' is not clear and better not be used, or may be replaced with something that can be understood by the field.

3. Page 7: "microduplication": there are more than simply microduplications of the Xq26 that cause XLAG. Replace with "genomic rearrangements that include microduplications and other defects"

4. The text in pages 6 and 7 is far too long

5. The text in the second half of page 16 is far too speculative; PKA compartmentalization has not been tested here and it is unclear how it links to what is being reported.

6. Activation of MAPK does not have to involve additional G-proteins; it may be seen as a consequence of Gs activation only, too.

Reviewer #2 (Remarks to the Author):

This study by Abboud et al. used an array of in vitro and in vivo animal models and human samples to attempt to identify the role of GPR101 in the GH axis. Based on their data, namely using a newly-developed mouse model with overexpression of GPR101 under the control of GHRHR promoter, they proposed that overexpression of the orphan GPCR constitutively activated both Gs and Gq/11 which would lead to enhanced GH secretion, but not proliferation of pituitary somatotrophs in mice up to the age of one year which displayed signatures of acromegalism. This paper is mainly technically sound. However I have many concerns about the interpretation of these data, in particular the analysis of the mouse model and how this work relates to the previous papers of the authors on this topic (A. Beckers in collaboration with CA Stratakis).

The generation of a mouse model with pituitary overexpression of GPR101 is an important step forward but in order to relate this to the aetiology of pituitary tumour formation in humans more characterisation of this model is required before publication. Specifically:

1/ The mouse model has been generated to recapitulate the overexpression of GPR101 in somatotrophs in X-LAG patients, however the evidence in Fig. 1A for FLAG-GPR101 expression in GH-expressing somatotrophs is not clear and a fuller description of the expression in all pituitary cell types is required. Additionally, the timing and level of expression, particularly foetal expression, is not reported. Given the early presentation of tumours in X-LAG patients, it is possible that expression at the onset of pituitary cell differentiation is required for tumour generation. Several transgenic lines were generated, was the phenotype in all these lines consistent? Was there variation in transgene expression level and how did this relate to the phenotype of animals?

2/ The increased circulating GH shown in Fig. 1B is consistent with the increased IGF-1 single-point detection of GH levels is far less informative than analysis of GH secretion profiles which are now easily detectable in mouse models using tail-tip blood sampling and ultra-sensitive mGH Elisa (initially developed by F Steyn in 2011 and widely used in many labs over the world). Moreover, as described years ago in human beings, GH pulses markedly decline in amplitude and regularity in mice older than 12-16-week-old. GH data from younger mice are therefore required and a fuller description of the growth pattern of transgenic mice is required, particularly at sexual maturation when body growth is highly dependent of GH pulsatility in mammals.

3/ The increased circulating PRL is consistent with that found in X-LAG patients but is hardly mentioned in the manuscript. Particularly pertinent on this point, the sex of animals used in the studies is not described and, given the sexual dimorphism in both GH and PRL secretion, an

important aspect that must be addressed before the manuscript would be acceptable for publication.

4/ There is no further information about GH-expressing somatotrophs in other figures like in Fig. 3 where expansion of the pituitary somatotroph population was evaluated. Given the focus of the study, was ki67 labelling increased in somatotrophs, which may simply be turning over faster in transgenic mice than controls. The GH gene expression and somatotroph GH content is also not described, which are important features given the increased GH expression in cell models described in Fig. 5F but lack of increased GH secretion in response to KCl shown in Fig. 3E.

5/ In Fig. 3E, ex vivo GH secretion responses (2/3- fold increase) to high/saturating GHRH stimulation can hardly recapitulate the x100/1000 fold increase in GH levels detected in vivo in both mouse models and humans.

6/ In Fig S3, I found that measuring GHRH and SST levels in systemic circulation was hardly relevant to the question of pituitary control since the pituitary portal system is a highly efficient system to transmit neurohormone pulses to pituitary targets while neurohormones are then largely diluted (upto x10,000-fold) into the systemic circulation. Additionally, interpretation of single time point measurements of GHRH given its patterned hypothalamic output is problematic. Indeed given the increased circulating GH and IGF-1, a decrease in circulating GHRH may have been expected.

7/ the assumption that overexpression of GPR101 under the control of GHRHr promoter would only be targeted to pituitary somatotrophs and lactotrophs was, in my view, too restrictive since GHRH receptors are known to be expressed in non-pituitary structures including hypothalamic neurons (e.g. glucose-sensing neurons, Stanley et al. Cell Metab 2013) which could be involved into the metabolic outputs reported in the present ms.

The in vitro characterisation of GPCR signalling from GPR101 overexpression is more impressive and has implications for understanding more general aspects of the regulation of pituitary cell proliferation, particularly with regard to pituitary tumours. However, there are alternative interpretations of the relationship between signalling and cell proliferation that would benefit from either further analysis and discussion. Specifically:

1/ Whilst the proposition that compartmentalisation of cAMP signalling leads to altered cell proliferation is one interpretation of the data show in Fig. 6, an alternative is that chronic cAMP signalling in response to overexpression of GPR101 gives a different response to acute activation which would occur in response to GHRH. This interpretation would also be consistent with the altered proliferation in response to FSK and 8-Br-cAMP. It is unclear whether Galphas siRNA treatment would have resulted in consistent, prolonged reduction in subunit expression or that this may have been transient, which again would have led to changes in cAMP signalling that may have differential effects on cell proliferation.

2/ The model proposed in Fig. 8 would be further supported by more experimentation of the combined effects of GPR101 and GHRH stimulation. Can the proliferative effects of GHRH in cells overexpressing GHRHR be blocked by GPR101? Increasing the GPR101 (or decreasing GHRHR/GHRH dose) would allow this analysis.

In conclusion the ms reports interesting data about signalling pathways downstream to GPR101 overexpression with, however, little additional information about the long-standing question of how GPR101 mutation leads to gigantism in children and acromegaly in adults and this mutation of an orphan GPCR lead to both increased release of GH and proliferation of GH-producing cells (Trivellin et al. NEJM 2014).

Reviewer #3 (Remarks to the Author):

This is an interesting and important manuscript that deals with GPR101 - an orphan GPCR that is overexpressed in X-linked agrotrophic gigantism (X-LAG), a severe form of pituitary gigantism that results in pituitary gland enlargement and tumor growth. The report investigates the mechanism by which GPR101 promotes gigantism creating and characterizing a GPR101 pituitary-specific overexpression mouse model. Studies reported in the manuscript establish role of GPR101 elevation in pituitary function in vivo as well as the signaling pathways responsible for the effects using in vitro signaling analysis in HEK293 and GH3 cells. Importantly, transgenic mice mimic many of the hallmarks of X-LAG, yet no proliferative/hyperplastic phenotypes are observed - a key

observation. The manuscript further establishes G protein coupling specificity of GPR101 leveraging its constitutive activity upon overexpression. The authors confirm that the increases in basal signaling in cAMP, IP1, and TGF- α assays are related to specific G protein pathways using a HEK293 G protein CRISPR approach.

This paper is well organized and the background is well explained. The data are overall convincing and the examination of the signaling link between the orphan GPCR GPR101 and its role in X-LAG is important not only to the understanding of the disease, but also to the basic pharmacology of the receptor.

Below is a list of a few recommendations for changes or other experiments:

1. An important control would be to determine if transgenic expression of GPR101 actually results in its overexpression and to document the extent of the overexpression. A western blot would be ideal, at least mRNA levels could be looked at. At the moment – it is not entirely clear if flag-tagging GPR101 creates signaling alterations or whether the receptor is indeed overexpressed.

2. There are a few issues related to data presented in Figure 4. Overall, having a more parallel structure would strengthen the conclusions, e.g. switching from shedding to IP1 assays and back while dropping some G proteins and arbitrary including others is not too logical. Perhaps Δ Gq/11 and Δ G12/13 could be examined in cAMP assays, Δ Gs in IP1 and a combination of G12/13/q/11 in both. It would be helpful to include a control using HEK293 cells without transiently transfected GPR101 and supplementing in each individual G protein to show the specific effect of GPR101 on TGF- α shedding. In Figure 4G,H– could the authors speculate why pERK was overall increased in the Δ Gtot condition even though there was no difference between No Receptor and GPR101 condition?

3. The evidence for G12/13 coupling is not strong. In the TGF- α release assay, deletion of Gq/11 or G12/13 had no impact on shedding. The follow-up to confirm the coupling of G12/13 to GPR101 was the TGF- α shedding assay in a G protein null (except Gi/o class) background, but a “no receptor” control was not performed. The data showing that G12/13 class signals downstream of GPR101 need to include this control. Another simple experiment to look at downstream G12/13 signaling would be to use the SRE-luciferase gene reporter assay and to measure basal signaling. Further, the only effect of G12/13 in GH3 cells was a mild decrease in cell proliferation when G12/13 was knocked down (Fig 6D). The authors make strong statements that GPR101 couples to G12/13, but G12/13 appears to play little to no role in most of the signaling pathways that the authors examined. The language discussing the role of G12/13 should be softened to reflect this and/or the authors should speculate in the discussion on the role of G12/13 signaling in GPR101 and elevated pituitary gland function. In fact, G12/13 data and conclusions could be dropped altogether without much of the detriment.

4. Data in Figure 6 and their interpretation may need further work. The authors note that co-transfection of GPR101 and GHRHR increased cAMP while not changing levels of proliferation. This is potentially a very interesting result. The authors initial hypothesis is that GPR101 on its own does not elicit sufficient changes in cAMP to alter proliferation. However, co-transfection of the two receptors significantly enhanced cAMP. The question is: can increased cAMP further enhance cell proliferation? The authors could perform a control experiment using GHRHR either with increasing agonist (if it is not already at a ceiling effect) or by increasing the receptor and showing that enhancing cAMP above what is shown in Fig 6C (~40 pmol/mL cAMP) will enhance proliferation and that there is something inherently different about the cAMP signal generated downstream of GPR101 that does not enhance proliferation. Looking at the effect of knocking down Gs in a cell co-transfected with both GHRHR and GPR101 on proliferation would also be of interest.

5. (Minor) The figure legend for 6F has 8-br-cAMP listed as an “adenylate cyclase activator”. Perhaps the authors meant to write a “PKA activator”.

Responses to Reviewers:

Reviewer #1:

Aboud et al describe a new mouse model (tg-GHRH-GPR101) and interesting new findings on how GPR101 may mediate its signaling. Unfortunately, the mouse data do not replicate the human disease (which is due to pituitary hyperplasia or tumor/adenoma formation) and the logic behind using the GHRH promoter to guide GPR101 expression to replicate the human condition is not very clear.

As a model, nevertheless, the data are interesting and the signaling findings are also of interest to the field.

We thank the reviewer for the positive comments. Our primary goal was not to recreate the disease in the transgenic mouse but to use this model to study and dissect the mechanisms associated with GPR101 function in the pituitary. We started with overexpression in somatotropes, as it is physiologically relevant, GPR101 being expressed normally in those cells (Trivellin et al. 2014, 2016). The selection of GHRHR as a promoter to drive expression of GPR101 in a sub-set of pituitary cells supports this goal as it allows for specific over-expression and provides insights on the contribution of GPR101 to normal pituitary function. While the mouse model did not faithfully replicate all aspects of X-LAG, it provides a very important *in vivo* demonstration that GPR101 is a strong facilitator of hormonal secretion in the pituitary that can lead to chronic effects of overgrowth and metabolic changes associated with GH hypersecretion. This lends support to the relevance of GPR101 signaling in physiological regulation of GH secretion and somatotrope activity, which is the true novelty of the reported findings.

Here are some additional specific comments:

Comment 1. There is no obvious strong GPR101 co-localization with GH (Fig. 1A) or Pit-1 (Suppl Fig. 1D). In the latter, GPR101 expression does not seem to be in the membrane as in the other figures (Fig. 1A and Suppl Fig. 1C). Better images for GPR101-Pit-1 have to be obtained.

Response: Although we observed co-localization between GH or Pit-1 and Flag-Gpr101, the images that were included were sub-optimal and did not illustrate this observation clearly. We acquired new images obtained with an improved protocol and have included them in the new figure 1 (**Fig. 1B & 1C**) and supplementary Figure 1 for the fetal Pit-1 colocalization (**Supp. Fig. S1I**). They are representative of several staining conditions of the anterior pituitary obtained in males and females.

Comment 2: Moreover, zoom-in boxes at higher magnification would be useful in all the figures.

Response: We have now included zoom-in boxes for each image. Also, we added arrows highlighting the important findings where appropriate.

Comment 3: The lack of clear GPR101-GH co-localization mimics what seen in patients with X-linked acroigantism (X-LAG). It is thus important to confidently show whether GPR101 is expressed in progenitors/undifferentiated cells, likely of the Pit-1 lineage, by showing better GPR101-Pit-1 co-localization as well as co-localization (or lack of) with other markers (e.g. Sox2, S100, etc).

Response: The pictures showing co-localization of Flag-Gpr101 and GH have been improved (**Fig. 1C**) and there are evident signs of expression of these two proteins within the same cells, somatotropes and somatomammotropes. We also now document more robustly the presence of Flag-Gpr101 in Pit-1-positive cells (**Fig. 1B** and **Supp Fig S1I** in the fetus). However, we could not demonstrate co-staining of Flag-Gpr101 with the progenitor marker Sox2 (**Supp Fig. S1J**), which indicates that the cells expressing the transgene are terminally differentiated (Andoniadou et al. 2013).

Pit-1 is a transcription factor that is necessary (but not sufficient) to differentiate progenitor into somatotropes, lactotropes and thyrotropes (Li et al. 1990). Our Pit-1 staining is consistent with this as not all Pit-1⁺ cells express Flag-Gpr101 (**Fig. 1B** and **Supp Fig S1I** in the fetus). As several comments from the Reviewers have noted prolactin (see below), we further documented the co-expression of Flag-

Gpr101 and prolactin (**Supp Fig. S1J**). In principle, terminally differentiated lactotrophs do not express *Ghrhr* but somatomammotropes in the anterior pituitary can secrete both GH and prolactin (Yeung et al. 2006). Thus, the observed hyperprolactinemia is likely due to the facilitation by Gpr101 of prolactin co-secretion with GH from somatomammotropes.

We have modified the text throughout to reflect these points and, in particular, regarding the choice of model we have rephrased in the introduction the section as follows:

"To better understand the place of GPR101 in somatotrope development and regulation, we developed a transgenic mouse model (*Ghrhr^{Gpr101}*) that expresses Gpr101 under the control of the *Ghrhr* promoter. This construction drives the expression of the transgene in the terminally differentiated somatotropes and somatomammotropes of the Pit-1 lineage^{20,21}."

Comment 4: GPR101 co-localization with GHRHR needs also to be tested to prove the cell type specificity of the rat *Ghrhr* promoter sequence used.

Response: We have included new images showing co-localization with *Ghrhr* (**Fig. 1A**) and the two receptors are expressed on the same cells (see inset).

Comment 5: GPR101 over-expression attained in the pituitary of TG mice (about 20 fold increase compared to controls) is lower than what seen in the tumors of X-LAG patients (up to hundreds-fold increase). This difference in magnitude could also be a factor explaining, maybe in part, the lack of pituitary tumorigenesis observed in the TG mice.

Response: The Reviewer makes a valid point. As compared with X-LAG tumor data from Trivellin *et al* (Trivellin et al. 2014), our mRNA quantities do not reach the same level. However, we would note that the two sets of data do not measure exactly the same thing. The human mRNA quantification looks at GPR101 total mRNA in pituitary tumors, whereas we (**Supp. Fig. S1E** in the new version) measure the amount of the transgene mRNA, not of the native Gpr101, which is unchanged (**Supp. Fig. S1D**). We have studied proliferation in a number of experiments and these all point in the same direction, that GPR101 is a facilitator of hormonal secretion but does not itself increase proliferation, even at increasing concentrations. This is in direct contrast to GHRH, which is a strong stimulator of proliferation, as reflected by our experiments and in keeping with ample fundamental and applied literature. Our new data also show that the overexpression of the transgene starts at the fetal stage (around E16.5, **Supp Fig. S1F & S1I**). During fetal, maturing and adult stages, despite overexpression of the transgene and the development of chronic hormonal hypersecretion, we found no evidence whatsoever of increased somatotrope proliferation. These *in vivo* observations are supported by the *in vitro* data on GH3 cells (**Fig. 6**). While we believe that the results we have obtained indicate that GPR101 in the pituitary is not strongly hyperproliferative, we have added a caveat regarding the potential role of much increased GPR101 expression in humans with XLAG promoting hyperplasia and tumorigenesis.

Comment 6: The results of mouse experiments (Fig. 1-3, and especially for Fig.1) should be broken down by sex. The authors do not specify if they analyzed both sexes or just males or females. If the numbers are small, experiments need to be expanded.

Response: We have revised the presentation of the experiments as requested. The phenotype that we observed was indeed similar between males and females and we studied a large group with a well-balanced number of animals from both sexes. We have revised the manuscript and figures to show results separately for males and females (See new panels of **Fig. 1-3**). Except for the prolactin (see below), there were no important differences between the sexes.

Comment 7: The authors somewhat ignore the fact that their mouse model got much more significant hyperPRLmia (Figure 1 D) than either GH or IGF1. In the text (page 8), they say "in addition to hyperPRLmia", but in fact hyperPRLmia was the main abnormality in addition to hyperGH...This finding is not adequately explained.

Comment 8: Was high PRL responsive to dopamine agonists?

Response to Comments 7 and 8: We thank the Reviewer for this observation and we recognize that we did not emphasize the hyperprolactinemia sufficiently in the original version. We have revised the manuscript to address prolactin secretion. It is well established that a population of cells that co-secrete GH and prolactin -somatomammotropes- exists in normal pituitary in mice, rats and humans. Also, somatomammotrope tumors in rats form the basis for the GH3 cell line that is a work-horse of pituitary research. In human, the involvement of somatomammotropes in pituitary tumors in patients with acromegaly and gigantism (including X-LAG) has been widely demonstrated. We identified pituitary cells that co-expressed the Flag-Gpr101 with prolactin and show these in the revised version (**Supp Fig. 1J**). As shown in the literature somatomammotropes have the secretory characteristics of both somatotropes and lactotropes and express GHRHR (Villalobos et al. 1997; Vidal et al. 2001; Seuntjens et al. 2002; Núñez et al. 2003; Deneff et al. 2005; Yeung et al. 2006; Ho et al. 2020). The hormonal profile of the transgenic animals also mirrors that seen in X-LAG where hyperprolactinemia is present in 39/40 cases in our clinical database and is responsive to dopamine agonists.

As noted by the Reviewer, the hyperprolactinemia is indeed a significant finding and another important piece of evidence of the role of GPR101 as a powerful modulator of pituitary hormone secretion.

In the analyses of the TG mouse pituitaries in the revised manuscript we now demonstrate

- That Gpr101 transgene shows substantial co-localization with Prolactin (**Supp. Fig. S1K**).
- The hyperprolactinemia is more pronounced in females (**Fig. 1F**)
- That the onset of hyperprolactinemia starts in young animals (**Fig. 1F**)
- That pituitary cells of the transgenic animals are still responsive to D₂ agonists (**Supp. Fig. S2F**)

These different elements have been included and discussed in the revised version of the manuscript.

Comment 9: In addition to body length, if available, the authors should also show whether there were any differences in tail length.

Response: We did not observe differences in tail length (**Supp. Fig. 2A**). This is consistent with published literature on growth in rodents where the tail is usually excluded from body length (nose-to-anus) measurements (Jewell and Fullagar 1966; Sagazio et al. 2008).

Comment 10: Growth curves for height and weight should be used in Fig. 1F-G in place of just one time point. In particular, mouse height shown in Fig. 1F, is not very convincing. 2 cluster of data can be seen. Do they represent males vs females or mice from different litters?

Response: We have revised the presentation of the data into curves as suggested. We now include in **Figure 1** a complete analysis of the growth curve for males and females (**Fig. 1G-I**). While there were no differences in young animals, the lengths diverged significantly at 24 weeks. The total weight was unaffected by the transgene addition (**Supp. Fig. 2D**).

Comment 11: RNA-seq data from mouse TG pituitaries would be very useful to confirm at a transcriptome pathway level the effects seen on signalling/proliferation/GH transcription (Fig. 4-6). Moreover, they can aid in the discovery of possible new pathways causing the observed phenotype.

Response: We thank the Reviewer for this insightful comment. We agree that such approaches are powerful to gain further understanding on signaling pathways. We have several ongoing projects focused on transcriptomic analysis (RNAseq, single cell RNAseq, qPCR, RNAscope,...) of different mouse models and organs where GPR101 produces its effects. The outcomes of such investigations go beyond the scope of the current study and we would prefer to address them at length and in the requisite detail in follow up studies later this year.

Comment 12: The results showing new signalling pathways activated by GPR101 are very interesting. Did the authors try to test the effect of the putative GPR101 ligand, GnRH(1-5)?

Response: GnRH(1-5) is a truncated version of the gonadotropic hormone GnRH and has been suggested as a ligand for GPR101 in Ishikawa cells, a human endometrial cancer cell line (Cho-Clark et al. 2014). We tried to recapitulate these findings in our assays but could not see any significant or specific signal using HEK293 cells (See **Rebuttal Figure 1** below). The reasons for these discrepancies are not clear at this point. We suggest that the difference in cell model and tissue may play a role in explaining this discrepancy. Indeed, we note that normal human endometrium has very low/absent GPR101 expression as RNA and protein (Human Protein Atlas), so the adenocarcinoma nature of the Ishikawa cell line may lead to receptor and signaling pathways profiles not seen in other models.

Rebuttal Figure 1. GnRH(1-5) fails to activate GPR101, in HEK293 cells.

Comment 13: Does GPR101 co-immunoprecipitate with the various G-proteins mentioned?

Response: We thank the Reviewer for this suggestion (and also Reviewer 3 for comments specifically about the coupling to $G_{12/13}$), and based on this we have undertaken new experiments which significantly strengthen the results presented. We performed a full array of co-immunoprecipitation studies with all individual G proteins and can confirm basal interaction with GPR101 for all G proteins tested except the G_i family (**Fig. 4I**). These data are consistent with those obtained in the other new cellular assays included in the revised Figure 4 (see below).

Comment 14: Could mathematical modelling (in silico) predict these non- G_s interactions?

Response: Some web-based and other platforms propose prediction of the G protein coupling for a given receptor using a variety of computational models and algorithms. However, generating a solid and informative dataset has always been a challenging task as it is still not clear what determines the preferential coupling of a receptor for a G protein. For instance, when the GPR101 sequence is inputted into the algorithm PRECOG (<http://precog.russelllab.org>) (Singh et al. 2019), the probability to couple to G_s is 0, in contrast to the experimental evidence. The software recognizes the probability of $G_{q/11}$ coupling but not to $G_{12/13}$ and surprisingly suggest the receptor may couple to $G_{i/o}$, which is not upheld experimentally. Another tool, PRED-COUPLE2 (<http://athina.biol.uoa.gr/bioinformatics/PRED-COUPLE2/>) (Sgourakis et al. 2005) returned a predicted coupling with all G proteins with G_s being the least probable.

Thus, these bioinformatic approaches are a useful complement, but the predictions obtained must be thoroughly validated in biochemical, cellular and in vivo studies.

Other minor comments:

1. Page 8: "...tibial length was markedly increased": delete "markedly"

- This has been corrected in the new version

2. Abstract and elsewhere: "prosecretory role" what is the meaning of this term? 'prosecretory' is not clear and better not be used, or may be replaced with something that can be understood by the field.

- We have edited the text to replace this with "promotes" or "facilitates" secretion.

3. Page 7: "microduplication": there are more than simply microduplications of the Xq26 that cause XLAG. Replace with "genomic rearrangements that include microduplications and other defects"

- All existing X-LAG cases (approx. 40 described and unpublished) are due to duplications at chromosome Xq26.3 that include *GPR101*. While there are a range of underlying genomic causes of this abnormality (microhomology mediated, fork stalling, Alu/Alu repeat mediated), all lead to duplications involving *GPR101*. We have adapted the text to state, "genomic rearrangements on chromosome Xq26.3 leading to duplications involving the *GPR101* gene".

4. The text in pages 6 and 7 is far too long

- The introduction has been reduced from ~740 words to ~570 words.

5. The text in the second half of page 16 is far too speculative; PKA compartmentalization has not been tested here and it is unclear how it links to what is being reported.

- We removed the speculative parts and replaced it by a more cautious statement:

" However, our results suggest that G_s activity in somatotropes does not invariably lead to proliferation and, via GPR101, may even counteract it. Divergent functional effects between different cAMP-elevating receptors in specialized cells has been documented for several decades. The results we obtained may be a manifestation of such compartmentalization of signaling but will require further investigation to be firmly demonstrated."

6. Activation of MAPK does not have to involve additional G-proteins; it may be seen as a consequence of G_s activation only, too.

- We agree with the reviewer on this point, this has been made clear in the manuscript.

Reviewer #2:

This study by Abboud et al. used an array of in vitro and in vivo animal models and human samples to attempt to identify the role of GPR101 in the GH axis. Based on their data, namely using a newly-developed mouse model with overexpression of GPR101 under the control of GHRHr promoter, they proposed that overexpression of the orphan GPCR constitutively activated both G_s and G_{q/11} which would lead to enhanced GH secretion, but not proliferation of pituitary somatotrophs in mice up to the age of one year which displayed signatures of acromegaly.

This paper is mainly technically sound. However I have many concerns about the interpretation of these data, in particular the analysis of the mouse model and how this work relates to the previous papers of the authors on this topic (A. Beckers in collaboration with CA Stratakis).

The generation of a mouse model with pituitary overexpression of GPR101 is an important step forward but in order to relate this to the aetiology of pituitary tumour formation in humans more characterisation of this model is required before publication.

We thank the reviewer for these positive comments.

Specifically:

Comment 1:

a) The mouse model has been generated to recapitulate the overexpression of GPR101 in somatotrophs in X-LAG patients, however the evidence in Fig. 1A for FLAG-GPR101 expression in GH-expressing somatotrophs is not clear and a fuller description of the expression in all pituitary cell types is required.

Response: We agree with the Reviewer on and we have improved the chosen images for the revised version based on an optimized protocol. These are included in the new version of the Figure 1 (particularly GH Fig. 1C). We have also added a full description of the co-staining in other pituitary cell sub-types (Supp. Fig. S1K-S1N).

b) Additionally, the timing and level of expression, particularly foetal expression, is not reported. Given the early presentation of tumours in X-LAG patients, it is possible that expression at the onset of pituitary cell differentiation is required for tumour generation.

Response: We performed additional staining and mRNA measurement of Flag-Gpr101 at the embryonic stage (Supp. Fig. S1F & S1H). We observed an expression of the transgene in E16.5 embryos. Thus, we can reasonably exclude that the absence of tumor is due to a late onset of Gpr101 overexpression.

c) Several transgenic lines were generated, was the phenotype in all these lines consistent?

Response: Other lines were generated and we confirmed the phenotype in another mouse line (called Tg2). Key data are shown in the new **Supplementary Figure S3**.

d) Was there variation in transgene expression level and how did this relate to the phenotype of animals?

Response: Tg2 incorporated less copies of the transgene (Supp. Fig. 3A). This translated into a trend of decreased FLAG-Gpr101 mRNA copies that did not reach statistical significance (Supp. Fig. 3B). There were no differences, however, in any of the hormonal, cellular or phenotypic characteristics of the two transgenic lines (see Supp. Fig. 3A).

Comment 2: The increased circulating GH shown in Fig. 1B is consistent with the increased IGF-1 single-point detection of GH levels is far less informative than analysis of GH secretion profiles which are now easily detectable in mouse models using tail-tip blood sampling and ultra-sensitive mGH Elisa (initially developed by F Steyn in 2011 and widely used in many labs over the world). Moreover, as described years ago in human beings, GH pulses markedly decline in amplitude and regularity in mice older than 12-16-week-old. GH data from younger mice are therefore required and a fuller description of the growth pattern of transgenic mice is required, particularly at sexual maturation when body growth is highly dependent of GH pulsatility in mammals.

Response: We agree with the Reviewer that our initial data lacked the time dimension. Therefore, we added full growth curves, for both males and females and included them in Fig. 1G & 1H. In addition, we included GH and IGF-1 determination at several time points (Fig. 1D & 1E).

Comment 3: The increased circulating PRL is consistent with that found in X-LAG patients but is hardly mentioned in the manuscript. Particularly pertinent on this point, the sex of animals used in the studies is not described and, given the sexual dimorphism in both GH and PRL secretion, an important aspect that must be addressed before the manuscript would be acceptable for publication.

Response: We recognize that the prolactin data were not given sufficient prominence in the original version of the manuscript. To address the comment of the Reviewer (and Reviewer #1), key data of Figures 1, 2 and 3 have been broken down by sex. We observed actually a difference between the prolactin levels in males and females, as could be expected (Fig. 1F). We included in the text additional discussion regarding hyperprolactinemia.

Comment 4:

A) There is no further information about GH-expressing somatotrophs in other figures like in Fig. 3 where expansion of the pituitary somatotroph population was evaluated. Given the focus of the study, was ki67 labelling increased in somatotrophs, which may simply be turning over faster in transgenic mice than controls.

Response: We studied the transgenic pituitaries and found few dividing cells and an identical Ki-67 ratio between transgenic and WT pituitaries (Fig. 3C of the revised version). Based on the Reviewer's comment, to exclude that the Ki-67 positive cells were specifically somatotropes, we performed co-staining with antibodies against GH and Ki-67 in immunofluorescence. These results were included in Figure 3D of the revised version. No Ki-67⁺ somatotropes were identified.

B) The GH gene expression and somatotroph GH content is also not described, which are important features given the increased GH expression in cell models described in Fig. 5F but lack of increased GH secretion in response to KCl shown in Fig. 3E.

Response: The GH gene expression in the pituitary was included in Supplementary Figure 2A of the original version of the article. However, we agree that this information is important and should have been better acknowledged and more prominently shown. Thus, we moved the panel showing GH mRNA to the main figures of the revised version (Fig. 3F). In addition, GH protein content (normalized to total protein) in the pituitaries of both males and females is now included (Fig. 3G). We also have revised the manuscript to show the pituitary weight, which was unchanged (Supp. Fig. S2G). These results indicate that, in addition to increased secretion (Fig. 3H & Fig. 5), the amount of GH in the somatotropes is increased.

The lack of differences in GH release between the KCl-stimulated pituitaries (Fig. 3H) has some possible explanation. Our goal was not to empty the stores of GH but to verify the condition of the cells. The concentration of KCl we used (30mM) is quite low compared to what is reported in similar experiments in the literature that stimulate with 100mM (Cochilla et al. 2000; Gaifullina et al. 2016). The infusion period was also shorter as we quickly washed-out the KCl.

Comment 5: In Fig. 3E, ex vivo GH secretion responses (2/3- fold increase) to high/saturating GHRH stimulation can hardly recapitulate the x100/1000 fold increase in GH levels detected in vivo in both mouse models and humans.

Response: The data in Fig. 1D are consistent (Between 2 and 3-fold at most) with the increase in the ex vivo model of the Fig. 3H (Fig. 3E in the initial version). In our opinion, the levels observed in humans cannot be really be compared with the transgenic mouse as they are the consequence of the activity of GH-secreting pituitary tumors that are not observed in the mouse model.

Comment 6: In Fig S3, I found that measuring GHRH and SST levels in systemic circulation was hardly relevant to the question of pituitary control since the pituitary portal system is a highly efficient system to transmit neurohormone pulses to pituitary targets while neurohormones are then largely diluted (upto x10,000-fold) into the systemic circulation. Additionally, interpretation of single time point measurements of GHRH given its patterned hypothalamic output is problematic. Indeed given the increased circulating GH and IGF-1, a decrease in circulating GHRH may have been expected.

Response: We fully recognize the point made by the Reviewer; study of GHRH and somatostatin dynamics outside of portal blood has to be treated cautiously; we included the data for the sake of completeness. The plasma levels of these hormone were measured as part of a general investigation where we monitored any change that may have occurred. We agree that they are not highly informative and little can be inferred. Thus, we removed mention of these data in the discussion.

Comment 7: the assumption that overexpression of GPR101 under the control of GHRHr promoter would only be targeted to pituitary somatotrophs and lactotrophs was, in my view, too restrictive since GHRH receptors are known to be expressed in non-pituitary structures including hypothalamic neurons (e.g. glucose-sensing neurons, Stanley et al. Cell Metab 2013) which could be involved into the metabolic outputs reported in the present ms.

Response: We agree that GHRHR could be expressed in other cell types. We did not find evidence of expression of Flag-Gpr101 in any of the principal brain structures (**Supp. Fig. S1H**). In addition, we performed immunofluorescent staining of the hypothalamus and did not see any neurons expressing the transgene (**Supp. Fig. 1G**). Although we cannot completely exclude some GHRHR promoter activity in the brain, it did not translate into measurable Flag-Gpr101 expression.

The in vitro characterisation of GPCR signalling from GPR101 overexpression is more impressive and has implications for understanding more general aspects of the regulation of pituitary cell proliferation, particularly with regard to pituitary tumours. However, there are alternative interpretations of the relationship between signalling and cell proliferation that would benefit from either further analysis and discussion.

Specifically:

Comment 1: Whilst the proposition that compartmentalisation of cAMP signalling leads to altered cell proliferation is one interpretation of the data shown in Fig. 6, an alternative is that chronic cAMP signalling in response to overexpression of GPR101 gives a different response to acute activation which would occur in response to GHRH. This interpretation would also be consistent with the altered proliferation in response to FSK and 8-Br-cAMP. It is unclear whether Galphas siRNA treatment would have resulted in consistent, prolonged reduction in subunit expression or that this may have been transient, which again would have led to changes in cAMP signalling that may have differential effects on cell proliferation.

Response: We thank the Reviewer for this insightful comment and agree that the compartmentalization explanation was too speculative and that other mechanisms may contribute to the observed absence of proliferation. The pattern, frequency and intensity of cAMP pulses could indeed have an impact on the cell response. Reviewer #3 had similar concerns on the compartmentalization concept that we raised. Hence, we have altered the strong statements on compartmentalization and included this alternative explanation of signaling dynamics suggested by the reviewer in the discussion:

"Other possible explanations exist for the differences observed in cellular response to GHRHR or GPR101 like the pattern of stimulation triggered by the two receptors. Indeed, when it is expressed, GPR101 activates the G_s continuously, in a chronic fashion, while GHRHR responds only to an acute stimulation by GHRH."

Comment 2: The model proposed in Fig. 8 would be further supported by more experimentation of the combined effects of GPR101 and GHRH stimulation. Can the proliferative effects of GHRH in cells overexpressing GHRHR be blocked by GPR101? Increasing the GPR101 (or decreasing GHRHR/GHRH dose) would allow this analysis.

Response: We thank the Reviewer for this constructive suggestion, which is echoed also by Reviewer 3. We have performed new experiments as suggested and the results are included in the new version of Figure 6. In panels **6D** and **6E**, we show that increasing the concentration of GPR101 potentiates cAMP increases driven by activated GHRHR, in a concentration-dependent manner. However, the amount of GPR101 had no measurable impact (potentiation or inhibition) on proliferation at any of the GHRH concentrations tested (**Fig. 6E**).

In conclusion the ms reports interesting data about signalling pathways downstream to GPR101 overexpression with, however, little additional information about the long-standing of how GPR101 mutation leads to gigantism in children and acromegaly in adults and this mutation of an orphan GPCR lead to both increased release of GH and proliferation of GH-producing cells (Trivellin et al. NEJM 2014).

Reviewer #3:

This is an interesting and important manuscript that deals with GPR101 - an orphan GPCR that is overexpressed in X-linked agrotrophism (X-LAG), a severe form of pituitary gigantism that results in pituitary gland enlargement and tumor growth. The report investigates the mechanism by which GPR101 promotes gigantism creating and characterizing a GPR101 pituitary-specific overexpression mouse model. Studies reported in the manuscript establish role of GPR101 elevation in pituitary function in vivo as well as the signaling pathways responsible for the effects using in vitro signaling analysis in HEK293 and GH3 cells. Importantly, transgenic mice mimic many of the hallmarks of X-LAG, yet no proliferative/hyperplastic phenotypes are observed – a key observation. The manuscript further establishes G protein coupling specificity of GPR101 leveraging its constitutive activity upon overexpression. The authors confirm that the increases in basal signaling in cAMP, IP1, and TGF- α assays are related to specific G protein pathways using a HEK293 G protein CRISPR approach.

This paper is well organized and the background is well explained. The data are overall convincing and the examination of the signaling link between the orphan GPCR GPR101 and its role in X-LAG is important not only to the understanding of the disease, but also to the basic pharmacology of the receptor.

We thank the reviewer for this encouraging comment.

Below is a list of a few recommendations for changes or other experiments:

Comment 1: An important control would be to determine if transgenic expression of GPR101 actually results in its overexpression and to document the extent of the overexpression. A western blot would be ideal, at least mRNA levels could be looked at. At the moment – it is not entirely clear if flag-tagging GPR101 creates signaling alterations or whether the receptor is indeed overexpressed.

Response: We thank the Reviewer for this good suggestion. The mRNA levels of both the wild type (unaffected) and the transgene are now included in **Supp. Fig. S1D & S1E**. A Western blot analysis of the presence of the native receptor is technically challenging because the available antibodies are not specific in our internal controls in the mouse. In terms of signaling, the Flag-Gpr101 behaves like the murine analog in cellular assays (constitutive cAMP increase). Thus, we hold that it is reasonable to infer that our observations mimic true receptor over-expression rather than signaling alterations.

Comment 2: There are a few issues related to data presented in Figure 4. Overall, having a more parallel structure would strengthen the conclusions, e.g. switching from shedding to IP1 assays and back while dropping some G proteins and arbitrary including others is not too logical.

Response: We thank the Reviewer for this excellent idea. We have now structured the presentation of the results to follow a more logical and simplified description assay by assay and have included several additional controls.

Perhaps

- Δ Gq/11 and Δ G12/13 could be examined in cAMP assays,

Response: The measures of cAMP in the Δ G_{q/11} cell lines were included in the supplementary information of the initial submission (Supp Fig. 4A). We have included cAMP determination in Δ G_{12/13} cell lines and moved those data in the Δ Gs Panel (now **Fig. 4A**). The MOCK conditions for these assays have also been included (**Supp. Fig. S5A**).

- Δ Gs in IP1 and a combination of G12/13/q/11 in both.

Response: We included additional control cell lines in the "IP₁" panel (**Fig. 4C**) as well as MOCK conditions (**Supp. Fig. S5C**). Cells lacking G12/13/q/11 are not currently available. As an alternative, we provide supplementary data on WT HEK293 cells transfected with GPR101 and treated with a

combination of siRNA against G_q, G₁₁, G₁₂ and G₁₃ and assayed for cAMP and IP₁ (**Supp. Fig. S5B & S5D**).

- It would be helpful to include a control using HEK293 cells without transiently transfected GPR101 and supplementing in each individual G protein to show the specific effect of GPR101 on TGF- α shedding.

Response: We now have included a comprehensive panel including the whole shedding experiment results (**Fig. 4H**). The transfection of individual G proteins in the absence of GPR101 in HEK293 cells had no impact on the basal signal.

- In Figure 4 G,H– could the authors speculate why pERK was overall increased in the ΔG_{tot} condition even though there was no difference between No Receptor and GPR101 condition?

Response: We noted this and were also intrigued by this increase. We reasoned at the time that the absence of G protein, except G_{i/o}, could disrupt the subtle equilibrium of basal signaling and promote signals originating from the constitutively activated G_{i/o}-coupled receptors, that are known to drive ERK phosphorylation, notably through the $\beta\gamma$ dimer (Wettschureck and Offermanns 2005). Although this is not strictly related to the present study, we wanted to verify this hypothesis. Thus, we treated the ΔG_{tot} cells with pertussis toxin (PTX) to blunt the G_{i/o} signaling in those cells. The increased p-ERK signal was clearly abolished, suggesting that G_{i/o} basal activation is elevated in the cell line (**Rebutal Fig. 2 & Supp. Fig. S5F**).

Rebuttal Figure 2. PTX treatment abolished p-ERK increase of the ΔG_{tot} line. **A.** WB Membrane stained with p-ERK and total ERK antibodies. Cell lysate were obtained from HEK. ΔG_{tot} cell lines transfected with GPR101 or empty vector (MOCK). **B.** Quantification by densitometry of four independent experiments. Results are expressed as mean \pm SEM, N=4.

Comment 3: The evidence for G_{12/13} coupling is not strong. In the TGF- α release assay, deletion of G_{q/11} or G_{12/13} had no impact on shedding. The follow-up to confirm the coupling of G_{12/13} to GPR101 was the TGF- α shedding assay in a G protein null (except G_{i/o} class) background, but a “no receptor” control was not performed. The data showing that G_{12/13} class signals downstream of GPR101 need to include this control.

Response: We agree with the Reviewer that the demonstration of the G_{12/13} coupling could have been more convincing. The shedding assay has the unique characteristic of “seeing” both G_{12/13} and G_{q/11}, so it seemed to us that when removing G_{q/11}, the remaining signal in shedding was likely to originate from

the G_{12/13} coupling. This was further confirmed with rescue experiment in a G protein null background (**Fig. 4H**).

Another simple experiment to look at downstream G_{12/13} signaling would be to use the SRE-luciferase gene reporter assay and to measure basal signaling. Further, the only effect of G_{12/13} in GH3 cells was a mild decrease in cell proliferation when G_{12/13} was knocked down (Fig 6D). The authors make strong statements that GPR101 couples to G_{12/13}, but G_{12/13} appears to play little to no role in most of the signaling pathways that the authors examined. The language discussing the role of G_{12/13} should be softened to reflect this and/or the authors should speculate in the discussion on the role of G_{12/13} signaling in GPR101 and elevated pituitary gland function. In fact, G_{12/13} data and conclusions could be dropped altogether without much of the detriment.

Response: We thank the Reviewer for such a constructive suggestion. While the SRE-luciferase is indeed a good option to measure activation of G_{12/13}, we selected a biochemical assay, the Rho-GTPase pull down (Nakaya et al. 2011) that convincingly shows activation of that pathway (**Fig. 4D**). Furthermore, we performed a complete G protein co-IP set that also shows evidence of G_{12/13} coupling (**Fig. 4I**). We agree that G_{12/13} impact on the GH3 response, at least for GH secretion and proliferation is limited. We could elect to remove the whole G_{12/13} dataset, but we feel that as we are comprehensively studying the G protein coupling profile of GPR101, we should include the information, even if it is not involved in GPR101-mediated GH secretion. Providing those results may facilitate and follow up projects focused on G_{12/13} in the field and we hope that the Reviewer accepts this approach.

Comment 4: Data in Figure 6 and their interpretation may need further work. The authors note that co-transfection of GPR101 and GHRHR increased cAMP while not changing levels of proliferation. This is potentially a very interesting result. The authors initial hypothesis is that GPR101 on its own does not elicit sufficient changes in cAMP to alter proliferation. However, co-transfection of the two receptors significantly enhanced cAMP. The question is: can increased cAMP further enhance cell proliferation? The authors could perform a control experiment using GHRHR either with increasing agonist (if it is not already at a ceiling effect) or by increasing the receptor and showing that enhancing cAMP above what is shown in Fig 6C (~40 pmol/mL cAMP) will enhance proliferation and that there is something inherently different about the cAMP signal generated downstream of GPR101 that does not enhance proliferation.

Response: We thank the Reviewer for this interesting insight. The data shown in Fig. 6C of the original version did not consider the impact of increasing concentrations of GHRH on the system. To comprehensively respond to this point, also noted by Reviewer 2, we undertook new experiments and now include two novel sets of data to study proliferation and cAMP in GH3 cells. We now show in revised **Fig. 6D** several GHRH concentration-response curves for different amounts of transfected GPR101. The potentiation of the GHRHR mediation of cAMP that was suggested by the original **Fig. 6C** now is more clearly seen as a leftward shift of the concentration-response curve reflecting a decrease in GHRH EC₅₀ (**Fig. 6D**). In parallel, we determined the proliferation of the GH3 cell population treated in the same manner. In revised **Fig. 6E**, we show that the potentiating effect does not translate into increased proliferation. Interestingly, the maximum level of cAMP is not modified by GPR101, as it is probably saturated. If the cAMP signal generated by GPR101 was identical in nature to that obtained after stimulation by GHRH, the proliferation should have been potentiated also.

Looking at the effect of knocking down Gs in a cell co-transfected with both GHRHR and GPR101 on proliferation would also be of interest.

Response: These control data have been included in **Fig. 6F**. They show that the decrease of proliferation in the absence of Gs is compensated when GPR101 is present. However, the expression of GPR101 does not inhibit GHRHR-mediated proliferation.

Comment 5: (Minor) The figure legend for 6F has 8-br-cAMP listed as an “adenylate cyclase activator”. Perhaps the authors meant to write a “PKA activator”.

Response: Yes, it was an error, and we have corrected it in the new version.

References

- Andoniadou CL, Matsushima D, Mousavy Gharavy SN, et al (2013) Sox2+ Stem/Progenitor Cells in the Adult Mouse Pituitary Support Organ Homeostasis and Have Tumor-Inducing Potential. *Cell Stem Cell* 13:433–445
- Cho-Clark M, Larco DO, Semsarzadeh NN, et al (2014) GnRH-(1-5) transactivates EGFR in Ishikawa human endometrial cells via an orphan G protein-coupled receptor. *Mol Endocrinol* 28:80–98
- Cochilla AJ, Angleson JK, Betz WJ (2000) Differential regulation of granule-to-granule and granule-to-plasma membrane fusion during secretion from rat pituitary lactotrophs. *J Cell Biol* 150:839–848
- Denef C, Pals K, Hauspie A, et al (2005) Combinatorial expression of phenotypes of different cell lineages in the rat and mouse pituitary. *Ann N Y Acad Sci* 1040:84–88
- Gaifullina AS, Yakovlev A V., Mustafina AN, et al (2016) Homocysteine augments BK channel activity and decreases exocytosis of secretory granules in rat GH3 cells. *FEBS Lett* 590:3375–3384
- Ho Y, Hu P, Peel MT, et al (2020) Single-cell transcriptomic analysis of adult mouse pituitary reveals sexual dimorphism and physiologic demand-induced cellular plasticity. *Protein Cell*. <https://doi.org/10.1007/s13238-020-00705-x>
- Jewell PA, Fullagar PJ (1966) Body measurements of small mammals: sources of error and anatomical changes. *J Zool* 150:501–509
- Li S, Crenshaw EB, Rawson EJ, et al (1990) Dwarf locus mutants lacking three pituitary cell types result from mutations in the POU-domain gene pit-1. *Nature* 347:528–533
- Nakaya M, Ohba M, Nishida M, Kurose H (2011) Determining the Activation of Rho as an Index of Receptor Coupling to G12/13 Proteins. In: *Methods in Molecular Biology*. pp 317–327
- Núñez L, Villalobos C, Senovilla L, García-Sancho J (2003) Multifunctional cells of mouse anterior pituitary reveal a striking sexual dimorphism. *J Physiol* 549:835–843
- Sagazio A, Xiao X, Wang Z, et al (2008) A single injection of double-stranded adeno-associated viral vector expressing GH normalizes growth in GH-deficient mice. *J Endocrinol* 196:79–88
- Seuntjens E, Hauspie A, Vankelecom H, Denef C (2002) Ontogeny of plurihormonal cells in the anterior pituitary of the mouse, as studied by means of hormone mRNA detection in single cells. *J Neuroendocrinol* 14:611–619
- Sgourakis NG, Bagos PG, Papasaikas PK, Hamodrakas SJ (2005) A method for the prediction of GPCRs coupling specificity to G-proteins using refined profile Hidden Markov Models. *BMC Bioinformatics* 6:1–12
- Singh G, Inoue A, Gutkind JS, et al (2019) PRECOG: PREdicting COupling probabilities of G-protein coupled receptors. *Nucleic Acids Res* 47:W395–W401
- Trivellin G, Bjelobaba I, Daly AF, et al (2016) Characterization of GPR101 transcript structure and expression patterns. *J Mol Endocrinol* 57:97–111
- Trivellin G, Daly AF, Faucz FR, et al (2014) Gigantism and acromegaly due to Xq26 microduplications and GPR101 mutation. *N Engl J Med* 371:2363–2374
- Vidal S, Horvath E, Kovacs K, et al (2001) Reversible transdifferentiation: Interconversion of somatotrophs and lactotrophs in pituitary hyperplasia. *Mod Pathol* 14:20–28

Villalobos C, Núñez L, Frawley LS, et al (1997) Multi-responsiveness of single anterior pituitary cells to hypothalamic-releasing hormones: A cellular basis for paradoxical secretion. *Proc Natl Acad Sci U S A* 94:14132–14137

Wettschureck N, Offermanns S (2005) Mammalian G Proteins and Their Cell Type Specific Functions. *Physiol Rev* 85:1159–1204

Yeung C-M, Chan C-B, Leung P-S, Cheng CHK (2006) Cells of the anterior pituitary. *Int J Biochem Cell Biol* 38:1441–1449

Reviewers' comments, second round:

Reviewer #1 (Remarks to the Author):

The revised manuscript addressed many of the concerns raised by the primary review. One issue is that remains is how much of the phenotype depends on high PRL levels. This reviewer understands that tempting to treat hyperprolactinemia will be large new experiment. But one could discuss this in the discussion.

Reviewer #2 (Remarks to the Author):

The authors are to be congratulated on the additional studies that have been performed that have addressed a range of issues that this and other reviewers had regarding the study and its interpretation. However, these additional studies and the responses to reviewers have only reinforced my view that the study has little relevance to progressing our understanding of the role of GPR101 in physiology or pathology of the GH axis in humans.

Original Comment 1(b): The relevance of the study to humans only lies in the study of the pathology associated with XLAG, since the cited studies of Trivellin (2014 and 2016) have shown that GPR101 is not expressed in the normal human somatotroph post-natally. To recapitulate the mis-expression of GPR101 in the somatotroph of XLAG patients, the model needs to express the protein at the same stages of pituitary development and with a comparable level. The authors have addressed the question of the stage of pituitary development (Supp. Fig. S1I) but not the extent of the over expression- my comment was asking for an assessment of whether the overexpression is comparable to that found in the somatotrophs of XLAG patients. This is acknowledged in the authors response to Reviewer 1, "the results we have obtained indicate that GPR101 in the pituitary is not strongly hyperproliferative, we have added a caveat regarding the potential role of much increased GPR101 expression in humans with XLAG promoting hyperplasia and tumorigenesis.". As stated in the rebuttal, the authors have shown a role for GPR101 in regulation of the mouse somatotroph in normal physiology but given it is not expressed in the post-natal human somatotroph (but is in the rodent), they are only addressing a role in rodent physiology and not human. If accepted for publication, the relevance to human pathology should be reduced dramatically, particularly in the abstract, and the differences in human and mouse somatotroph biology and potential species differences in the proliferative response of the cells to GPR101 expression highlighted.

Original Comment 2: The additional data of plasma GH and IGF-1 at various ages is welcome and but the authors have not addressed the point being made. Single time-point measurements of circulating GH give very limited information and understanding of the effects on secretion require multiple measurements over a period of hours, since the pattern of GH secretion is as important as its level. Whilst it is not uncommon to report single time-point measurements as opposed to profiles this is not informative and in some cases can be misleading.

Original Comment 5: This comment was not made regarding pituitary tumours in humans, as suggested in the rebuttal, but addresses the relevance of measuring secretory output ex vivo. When measured in vivo, GHRH elicits a 100-1000 fold increase in GH secretion which is not recapitulated in ex vivo experiments. Thus, the relevance of the increase in an ex vivo response of 2-3 fold is questionable. Whilst it is consistent with the increased circulating GH shown in 1D, the comment above explains why the data in 1D may also be misleading and the relevance of both to in vivo physiology is questionable.

Other comments have been answered by either inclusion of additional data or alteration to the text to my satisfaction.

Reviewer #3 (Remarks to the Author):

the authors have addressed all my concerns. Great job revising!!!

Responses to Reviewers:

Reviewer #1:

The revised manuscript addressed many of the concerns raised by the primary review. One issue is that remains is how much of the phenotype depends on high PRL levels. This reviewer understands that tempting to treat hyperprolactinemia will be large new experiment. But one could discuss this in the discussion.

Response: We thank the Reviewer for the comment and we agree that further investigation of the hyperprolactinemia will benefit from detailed specific research. We have added text to the revised version to acknowledge this point:

“The hyperprolactinemia that is also encountered in the *Ghrhr^{Gpr101}* mice requires specific studies to determine the precise mechanisms by which PRL dysregulation occurs and how this impacts the phenotype of these animals.”

Reviewer #2:

The authors are to be congratulated on the additional studies that have been performed that have addressed a range of issues that this and other reviewers had regarding the study and its interpretation. However, these additional studies and the responses to reviewers have only reinforced my view that the study has little relevance to progressing our understanding of the role of GPR101 in physiology or pathology of the GH axis in humans.

Original Comment 1(b): The relevance of the study to humans only lies in the study of the pathology associated with XLAG, since the cited studies of Trivellin (2014 and 2016) have shown that GPR101 is not expressed in the normal human somatotroph post-natally. To recapitulate the mis-expression of GPR101 in the somatotroph of XLAG patients, the model needs to express the protein at the same stages of pituitary development and with a comparable level. The authors have addressed the question of the stage of pituitary development (Supp. Fig. S11) but not the extent of the over expression- my comment was asking for an assessment of whether the overexpression is comparable to that found in the somatotrophs of XLAG patients. This is acknowledged in the authors response to Reviewer 1, "the results we have obtained indicate that GPR101 in the pituitary is not strongly hyperproliferative, we have added a caveat regarding the potential role of much increased GPR101 expression in humans with XLAG promoting hyperplasia and tumorigenesis.". As stated in the rebuttal, the authors have shown a role for GPR101 in regulation of the mouse somatotroph in normal physiology but given it is not expressed in the post-natal human somatotroph (but is in the rodent), they are only addressing a role in rodent physiology and not human.

If accepted for publication, the relevance to human pathology should be reduced dramatically, particularly in the abstract, and the differences in human and mouse somatotroph biology and potential species differences in the proliferative response of the cells to GPR101 expression highlighted.

Response: We thank the Reviewer for the many detailed and significant comments raised during review of our original manuscript and we have endeavored to address these points with experimental data and revised argumentation. We believe that the findings on signaling relating to GPR101 in the somatotrope axis form the basis for exploring and defining its precise role in different species, at different periods of pituitary development and in clinical scenarios. We understand the concern that our results mainly relate to the cell and animal models chosen and that our clinically-based conclusions should be more cautious. This point has been echoed by the Editor, and we have, therefore, reduced the inferences between our results and X-LAG in the Introduction, results and Discussion. The Abstract has also been extensively revised to address this issue. Thus, we focus the scope of the manuscript on signaling based findings. We hope that these revisions to the manuscript will meet with the Reviewer's approval.

Original Comment 2: The additional data of plasma GH and IGF-1 at various ages is welcome and but the authors have not addressed the point being made. Single time-point measurements of circulating GH give very limited information and understanding of the effects on secretion require multiple measurements over a period of hours, since the pattern of GH secretion is as important as its level. Whilst it is not uncommon to report single time-point measurements as opposed to profiles this is not informative and in some cases can be misleading.

Response: We fully acknowledge that the normal and pathological secretion of GH is a complex issue, mainly due to the pulsatility patterns that can vary diurnally or during developmental phases. We agree that this is an aspect that remains to be explored in these animals and other models and we have added specific text to the Discussion in the revision to highlight the Reviewer's point:

“Further aspects of Gpr101-related hormonal secretion in mice remain to be explored, such as, the important issue of potential alterations in GH pulsatility.”

Original Comment 5: This comment was not made regarding pituitary tumours in humans, as suggested in the rebuttal, but addresses the relevance of measuring secretory output ex vivo. When measured in vivo, GHRH elicits a 100-1000 fold increase in GH secretion which is not recapitulated in ex vivo experiments. Thus, the relevance of the increase in an ex vivo response of 2-3 fold is questionable. Whilst it is consistent with the increased circulating GH shown in 1D, the comment above explains why the data in 1D may also be misleading and the relevance of both to in vivo physiology is questionable.

Response: The Reviewer highlights an important distinction between in vivo and ex vivo responses to GHRH. Our experimental goal with the ex vivo experiment was to demonstrate differences in responses to GHRH and GH output between the two mouse lines. Hence, in the experimental design, the pituitary tissue was only acutely treated with GHRH and we did not aim to reach the maximum possible GH secretion. In addition, the physiology of an extracted, isolated organ suspended in a buffer is necessarily impacted by the procedure. Therefore, we acknowledge that we should have added a note of caution. We have now included a caveat about in vivo and ex vivo GHRH responses in the revised Discussion:

“Similarly, the magnitude of the secretory responses to GHRH stimulation seen in our ex vivo experiments of pituitary tissue need to be balanced against the greater magnitude of GH responses to GHRH that occur in vivo.”

Other comments have been answered by either inclusion of additional data or alteration to the text to my satisfaction.

Response: Again, we thank the Reviewer for the very thorough review and the many knowledgeable insights given; our work has been improved significantly by addressing the constructive critiques.

Reviewer #3

The authors have addressed all my concerns. Great job revising!!!

Response: We thank the Reviewer for the very positive and kind comment.